# NEXT STATE PREDICTION GIVES RISE TO ENTANGLED, YET COMPOSITIONAL REPRESENTATIONS OF OBJECTS

## ABSTRACT

Compositional representations are thought to enable humans to generalize across combinatorially vast state spaces. Models with learnable object slots, which encode information about objects in separate latent codes, have shown promise for this type of generalization but rely on strong architectural priors. Models with distributed representations, on the other hand, use overlapping, potentially entangled neural codes, and their ability to support compositional generalization remains underexplored. In this paper we examine whether distributed models can develop linearly separable representations of objects, like slotted models, through unsupervised training on videos of object interactions. We show that, surprisingly, models with distributed representations often match or outperform models with object slots in the tasks they were trained to perform. Furthermore, we find that linearly separable object representations can emerge without object-centric priors, with auxiliary objectives like next-state prediction playing a key role. Finally, we observe that distributed models' object representations are never fully disentangled, even if they are linearly separable: Multiple objects can be encoded through partially overlapping neural populations while still being highly separable with a linear classifier. We hypothesize that maintaining partially shared codes enables distributed models to better compress object dynamics, potentially enhancing generalization.

## 1 INTRODUCTION

Humans naturally decompose scenes, events and processes in terms of the objects that feature in them (Tenenbaum et al., 2011; Lake et al., 2017). These object-centric construals have been argued to explain humans' ability to reason and generalize successfully (Goodman et al., 2008; Lake et al., 2015; Schulze Buschoff et al., 2023). It has therefore long been a chief aim in machine learning research to design models and agents that learn to represent the world compositionally, e.g. in terms of the building blocks that compose it. In computer vision, models with *object slots* learn to encode scenes into a latent, compositional code, where each object in the scene is modelled by a distinct part of the latent space. This strong architectural assumption allows the models to learn representations that improve compositional generalization (Brady et al., 2023; Wiedemer et al., 2023) and reasoning about objects (Wu et al., 2022).

Slotted representations are often contrasted with distributed representations. Models with distributed representations encode information about a scene, and potentially the objects that compose it, in overlapping populations of neurons. Can models with distributed representations learn to encode objects in a compositional way without supervision? And can distributed coding schemes offer advantages over *purely* object-centric coding schemes?

By compressing properties of multiple objects in a shared code, models with distributed representations could potentially gain richer representations where scene similarities are more abundant (Smola & Schölkopf, 1998; Lucas et al., 2015; Demircan et al., 2023; Garvert et al., 2023). For instance, if two objects are represented similarly, the model could use what it knows about the dynamics of one object to generalize about the dynamics of the other object. This could in turn facilitate learning, potentially at the loss of fully separable object representations.

In our study, we offer experimental evidence that **models with distributed representations can learn compositional construals of objects** in an unsupervised manner, when trained on sufficiently

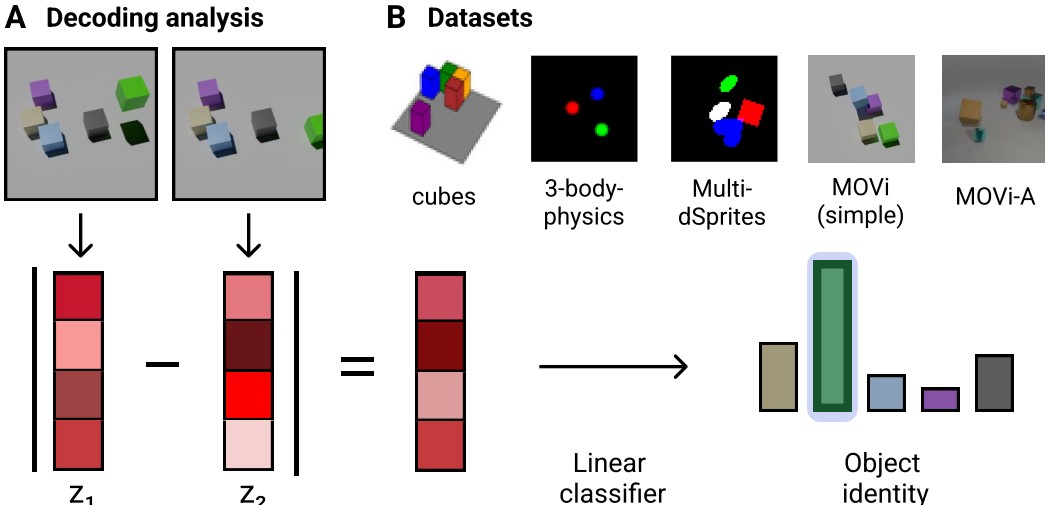

Figure 1: Overview of the decoding analysis and datasets. **A**: We propose a simple test for assessing compositional object representations. After unsupervised pre-training on object videos, we evaluate the linear separability of models' latent object representations. This is done by training a linear classifier on the absolute differences of two successive encoded frames where only one object changes. **B**: We evaluate the models on five datasets of dynamically interacting objects, ranging from simple depictions of blocks and sprites to realistic simulations of 3D objects.

large datasets. Across 5 datasets that consist of dynamically interacting objects we see that models with distributed representations either match or outperform their slot-based counterparts in the tasks they were trained to perform (image reconstruction or dynamics prediction). Next, we define a simple metric inspired by Higgins et al. (2016) that quantifies how accurately object identities can be linearly decoded from a model's latent representations (see Figure 1). We see that as training data size increases, the models with distributed representations develop gradually more disentangled representations of objects. However, while object representations become separable, their properties remain encoded through *partially* overlapping populations of neurons, potentially allowing for richer generalization. Investigating the effect of training objective and loss function on the separability metric, we find that next-state prediction is a crucial component for the development of separable object representations for models without object slots.

While we see object disentanglement emerge without any supervision or regularization, this disentanglement is not absolute. Distributed models represent object properties in partially shared latent spaces. We speculate that this can facilitate generalization: When comparing models' representations of object dynamics, we see clear clustering based on object identity (indicating separable object representations), but also clustering based on what type of transformation was applied to the object. This means that transformations such as object rotations, object scaling or object movements are represented more similarly, independently of which object they are applied to. Such compositional codes for group transformations of objects are made possible by the fact that all objects share a common latent space, instead of occupying separate ones, suggesting that there are benefits to distributed coding schemes.

## 2 RELATED WORK

Object-centric representations have been argued to improve the sample efficiency and generalization ability of vision and dynamics models in compositional domains (Elsayed et al., 2022; Kipf et al., 2019; Wiedemer et al., 2023; Wu et al., 2022; Locatello et al., 2020b). Object-centric representations, given an appropriate model architecture, can also be learned in an unsupervised manner (Kipf et al., 2019; Locatello et al., 2020b; Brady et al., 2023) and on real-world datasets (Seitzer et al., 2022). Previous studies have highlighted the importance of architectural features, like object-

slots (Greff et al., 2020; 2019; Dittadi et al., 2021), as well as data properties, like having access to temporal information (Zadaianchuk et al., 2024).

Other lines for learning compositional representations have been proposed as well. Atzmon et al. (2016) and Johnson et al. (2017) introduced datasets to test compositional generalization in machine learning models. While we focus on slot-based models, there are different approaches to learning object-centric representations. Patch-based models such as SPACE (Lin et al., 2020) and MarioNette Smirnov et al. (2021) decompose scenes into disentangled representations by reconstructing the input from patches. Keypoint-based models such as DLP (Daniel & Tamar, 2022) build on representations as sets of geometrical points as an alternative to single-vector representations.

There is a considerable overlap in the literature on object-centric and disentangled representations. A disentangled representation is one "which separates the factors of variation, explicitly representing the important attributes of the data" (Bengio et al., 2013; Locatello et al., 2020a; Higgins et al., 2016; 2018). In a disentangled representation, a change in a single ground truth factor should lead to a change in a single factor in the learned representation (Locatello et al., 2019; Ridgeway & Mozer, 2018; Kim & Mnih, 2018). Information bottlenecks methods like $\beta$-VAEs have also been shown to be able to disentangle object features (Burgess et al., 2018; Higgins et al., 2017).

In recent work, Brady et al. (2023) put forward a measure of representational object-centricness that measures "if there exists an invertible function between each ground-truth slot and exactly one inferred latent slot". We work with a related metric suitable for generic model classes (e.g. without image decoders) that instead measures the degree to which changes to individual objects can be predicted from changes in a model's latent representations.

## 3 METHODS

### 3.1 MODELS

We focus on models that learn representations of scenes in an unsupervised manner, e.g. without information about object identities provided as labels or masks. Unsupervised training regimes, such as auto-encoding (Kingma, 2013), denoising (He et al., 2022) and contrastive objectives (Chen et al., 2020) have shown promise as representation learning tools in many domains, ranging from image and language understanding (Radford et al., 2021) to reinforcement learning (Schwarzer et al., 2020; Gelada et al., 2019; Saanum et al., 2024). In this paper we investigate two classes of such training objectives: *i*) Reconstruction-based or auto-encoding objectives, where the goal is to encode and reconstruct images of scenes of objects. And *ii*), contrastive objectives, where the goal is to maximize embedding similarities of positive pairs and minimize similarities of negative pairs. Accordingly, the models rely on an image encoder, a Convolutional Neural Net (CNN) in our case, to map images of objects to latent representations. For auto-encoding models we additionally equip the model with a CNN decoder that maps the latent representation back to pixel-space.

$$z_t = e_\theta(x_t) \tag{1}$$
$$\tilde{x}_t = g_\theta(z_t) \tag{2}$$

Here $z_t$ is the model's representation, and $e_\theta$ and $g_\theta$ are the CNN encoder and decoder. $x_t$ and $\tilde{x}_t$ are the image and reconstruction, respectively. We subscript image and representation variables with the time-point $t$ since our data are dynamic. Having access to this temporal information about the data, we also consider models that use future-state prediction as an auxiliary objective for representation learning. Observing how objects interact dynamically can provide the models with useful cues about object identities, and could facilitate learning systematic representations of objects (Zadaianchuk et al., 2024). When modelling the dynamics of the object data, we equip the model with a latent dynamics module that predicts the model's representation at the next time point $t + 1$, given the current representation, and an action $a_t$, if the dynamics data is accompanied by actions.

$$\tilde{z}_{t+1} = d_\theta(z_t, a_t) \tag{3}$$

Here $d_\theta$ denotes the dynamics module, which is a Multi-Layer Perceptron (MLP) in the case that the dynamics are Markovian, e.g. fully predictable from the information provided in the current

observation $x_t$ (and potentially action $a_t$). In non-Markovian settings we use a causal Transformer that integrates information across representations of past observations $(z_{t-n}, ..., z_{t-1}, z_t)$ to predict the dynamics. See Appendix B for details on the model architecture and hyperparameters.

For the auto-encoding models, we train the encoder and decoder to reconstruct the *current* frame from the current representation in the static setting, and the *next* frame from the predicted *next* latent representation in the dynamic setting. Here, we additionally train the dynamics model to minimize the distance between the predicted and actual representation of the next frame. This leaves us with the following loss functions:

$$\mathcal{L}_{\text{AE-static}} = ||x_t - g_\theta(z_t)||_2^2 \tag{4}$$

$$\mathcal{L}_{\text{AE-dynamic}} = ||x_{t+1} - g_\theta(\tilde{z}_{t+1})||_2^2 + ||z_{t+1} - d_\theta(z_t, a_t)||_2^2 \tag{5}$$

We refer to these models as the *auto-encoder* and *sequential auto-encoder*, respectively.

For the contrastive models, we consider both a static and dynamic training scheme as well. In the static case, we present the model with a frame $x_t$ as well as a randomly augmented view of the same frame $h(x_t)$ (Laskin et al., 2020; Grill et al., 2020). The model is then trained to minimize the embedding distance between the original and augmented view of the image, while maximizing the embedding distance between the original image and its representations of augmented views of other frames $x^-$ in the batch, up to a margin $\lambda$. In the dynamic setting we train the contrastive model as follows: Given an initial latent representation (and potentially action), we train the encoder and dynamics model to produce a prediction that is as close as possible to the encoded representation of the next frame $z_{t+1}$, and that is maximally far away from encoded representations of other frames $z^-$, up to a margin $\lambda$. The loss functions take the following form:

$$\mathcal{L}_{\text{contrastive-static}} = ||z_t - e_\theta(h(x_t))||_2^2 + \max(0, \lambda - ||e_\theta(h(x^-)) - z_t||_2^2) \tag{6}$$

$$\mathcal{L}_{\text{contrastive-dynamic}} = ||z_{t+1} - d_\theta(z_t, a_t)||_2^2 + \max(0, \lambda - ||z^- - d_\theta(z_t, a_t)||_2^2) \tag{7}$$

We refer to the static contrastive model as *CRL*, for Contrastive Representation Learner, and the dynamic contrastive model as *CWM*, for Contrastive World Model.

### 3.1.1 SLOTTED MODELS

We compare the auto-encoding models and CWM to baselines which attempt to learn slotted representations. As a baseline to the auto-encoding models, we implement Slot Attention (Locatello et al., 2020b), an auto-encoder that reconstructs images as an additive composition of multiple object slots. The slots compete to represent the objects in the scene using an iterative attention mechanism, and the full model is trained with a simple auto-encoding objective as in equation 5.

The contrastive dynamics model is compared to a *structured* variant, the Contrastive Structured World Model (CSWM) (Kipf et al., 2019), that decomposes the scene into distinct object slots, and uses a graph neural network to predict how these object slots evolve over time. In non-Markovian settings, we replace the graph neural network with a Transformer encoder that applies spatio-temporal attention over a sequence of past object-slot representations, akin to the Slotformer architecture (Wu et al., 2022). Here too, the loss function exactly matches the one used to train the contrastive dynamics model with distributed representations.

### 3.2 ASSESSING OBJECT REPRESENTATIONS

How can we quantify the degree to which a non-slotted model has learned systematic object representations? While many metrics are possible, we propose one which is both simple and has connections to other metrics proposed to quantify representation disentanglement. If a representation of an object $o_i$ is disentangled from representations of other objects $o'$, then a change to $o_i$ should only change one subspace of the models' latent representation $z$. Additionally, this subspace should not be affected by changes to any of the other objects $o'$. In other words, each object is represented across completely non-overlapping populations of neurons. Complete disentanglement is a tall order

for models without the structural properties of slotted models. To get a continuous relaxation of this absolute object disentanglement metric we ask a related question of the models' latent spaces: Given a set of changes to individual objects in a scene, how accurately can a linear classifier predict which object was changed from the resulting absolute difference in the model's latent representations? The accuracy of this linear classifier on an evaluation set is our proposed metric. This metric is in fact a variation of the disentanglement metric proposed in Higgins et al. (2016), but applied to objects rather than ground truth generative features. Even if a model attains a perfect score on this metric, it does not necessarily mean that it represent objects in perfectly disjoint, non-overlapping populations of neurons. To illustrate, if a change to object $o_i$ always changes latent $z^1$ marginally and latent $z^2$ greatly, and a change to object $o_j$ changes $z^1$ greatly and $z^2$ marginally, a linear classifier can reliably separate the two objects in terms of the change in $z$, despite them having entangled representations. In other words, it is possible to use overlapping neural codes to represent objects, while still having object representations that are linearly separable. We investigate this further in Section 5.

In practice, we implement our metric by constructing datasets consisting of pairs of images $(x_t^i, x_{t+1}^i)$ from the data domain on which a model was trained. The only difference between these two images is that a single object has changed from time $t$ to $t + 1$. For each such pair we associate it with a label $y^i$, a categorical variable indicating *which* object was altered from $t$ to $t + 1$. We then extract a model's representation of each frame in the pair (after it has been trained), and compute the vector of absolute differences between these two representations:

$$|\Delta^i| = |e_\theta(x_t^i) - e_\theta(x_{t+1}^i)| \qquad (8)$$

From the set of ensuing absolute difference vectors $\mathcal{X} = (|\Delta^1|, ..., |\Delta^n|)$ we train a linear classifier to predict the corresponding object labels $\mathcal{Y} = (y^1, ..., y^n)$ while minimizing the $L_1$ norm of the learned coefficients, as recommended by Higgins et al. (2016). We report the accuracy on a left out subset of $(\mathcal{X}, \mathcal{Y})$ that the classifier was not trained on.

## 4 EXPERIMENTS

We trained the two classes of distributed models, as well as their object-centric counterparts, on five datasets of dynamically interacting objects. Two of these datasets, cubes and 3-body physics, were introduced in Kipf et al. (2019) to showcase how object-centric representations facilitate learning of object dynamics. Extending the evaluation, we created our own dataset of object interactions based on the dSprites environment (Matthey et al., 2017). This dataset consisted of four sprites with different shapes and colors traversing latent generative factors, such as $(x, y)$-coordinates, scale and orientation, on a random walk. Lastly, we trained our models on two more complex Multi-Object Video (MOVi) datasets generated using the Kubric simulator (Greff et al., 2022), a 3D physics engine for simulating realistic object interactions. We generated one dataset consisting of $14,000$ videos with a constant set of five cubes with fixed physical properties that interacted (initial object conditions such as directional velocities and position were randomized for each video). We refer to this dataset as MOVi (simple), due to the constant object properties. Additionally we trained models on the MOVi-A dataset, consisting of almost $10,000$ videos where the number of objects, their shapes and physical properties, such as mass and friction, are not fixed and vary across videos. The cubes and Multi-dSprites datasets had action variables that accompanied the videos, and were predictive of the way the objects would change from one frame to another. The other datasets were action-free. All models were trained for 100 epochs with five random seeds on the cubes, 3-body physics and multi-dSprites datasets, and for 125 epochs with three random seeds on the MOVi datasets. Furthermore, to assess the effect of dataset size on our evaluation metrics, we split each dataset up in different sizes.

We evaluated the slotted CSWM and distributed CWM models' prediction abilities by measuring the accuracy with which they could predict novel object trajectories of length $n$ from an unseen evaluation set in an open loop manner. Prediction accuracy was estimated as the percentage of test trajectories where the predicted latent state $\tilde{z}_{t+n}$ at the end of the trajectory was closest in terms of Euclidean distance to the model's representation of the last frame in the trajectory $z_{t+n}$ out of 1000 evaluation trajectories. In other words, a prediction is deemed correct if the final encoded

state is the closest in $L_2$ distance to the predicted final state in the corresponding video, and incorrect if it is closer to any of the other 999 predictions. For the `cubes`, `3-body physics` and `Multi-dSprites` datasets, we conducted the evaluations with a trajectory length of $n = 10$, and a trajectory length of $n = 3$ in the more complex MOVi datasets.

To assess object-separability we created evaluation videos for all five datasets. In these evaluation sets only single objects from the respective object domain were changed while all other objects in the scene remained fixed. After training models on each of the datasets, we assessed how well one could linearly classify which object had moved using the protocol described in Section 3.2.

### 4.1 Object slots are not necessary for learning object dynamics

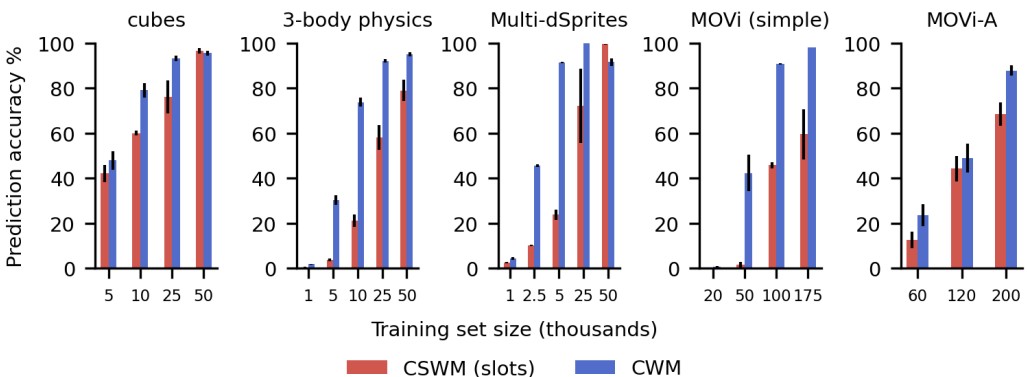

Figure 2: Prediction accuracies for slotted and non-slotted contrastive dynamics models. In all five datasets we see that the CWM is not only competitive, but sometimes outperforms the CSWM when it comes to predicting object dynamics. Scores are averaged over five seeds, with error bars depicting the standard error of the mean.

Evaluating the prediction accuracy of the CSWM and CWM, we observe that object-slots are not necessary for accurately predicting object dynamics. In fact, the CWM models often outperformed their slotted counterparts (see Figure 2). As expected, we see test accuracy generally increase with training set size. This suggests that compositional generalization about objects, the ability to generalize about properties of objects in novel constellations and combinations, does not require explicit object-centric priors as provided by slotted architectures.

We further tested the models' compositional generalization ability by designing datasets with systematic deviations in training and test data. We constructed two new datasets composed of 3000 videos each in the MOVi environment. The first dataset (`Novel objects`) consists a training set of videos with one to four spheres, and a test set with five to eight spheres. The second dataset consists of a training set of two red cubes and two green spheres, and a test set of two green cubes and two red spheres (`Color swap`). In both experiments we see that there's a train-test disparity that gradually diminishes with more and more training data for both CSWM and CWM (see Fig. 8). CWM retains an edge in sample-efficiency

#### 4.1.1 Down-stream tasks

Lastly we assessed the performance of CSWM and CWM in downstream tasks. In the first one, a control task, we train a Soft Actor Critic (SAC) Haarnoja et al. (2018) agent to manipulate a randomly sampled sprite in the Multi-dSprite environment to go to a particular location on the grid. The SAC agent receives observations that are the encodings of the scene produced by one of the pretrained models. We evaluate the agent using the embeddings of CSWM, CWM and the autoencoder, and see that the agent trained to perform control using CSWM representations performs the best, with the CWM-based agent trailing closely behind (see Fig. 9). This suggests that slotted representation could offer advantages in downstream control tasks. Indeed, the lower the object-

separability score, the worse the downstream control performance. Having object-slots could present a significant advantage in these settings, as it facilitates object-centric learning.

In the second downstream task we used the representations of the trained MOVi-A models to predict a novel quantity. We froze the encoders of CSWM and CWM and trained a linear classifier to predict the number of objects present in a scene. We constructed three datasets, where the number of possible objects present in a scene increased from two to four. In the first dataset there were therefore two possible labels (does the scene contain one or two objects?), and in the last dataset four labels (does the scene contain one, two, three or four objects). Here we see that both models can predict object cardinality better than chance with a simple linear classifier with only minor differences in prediction accuracy between them (see Fig 10). Moreover, prediction accuracy generally increases the more data the models were trained with in their original task.

## 4.2 PREDICTING OBJECT DYNAMICS IMPROVES OBJECT SEPARABILITY

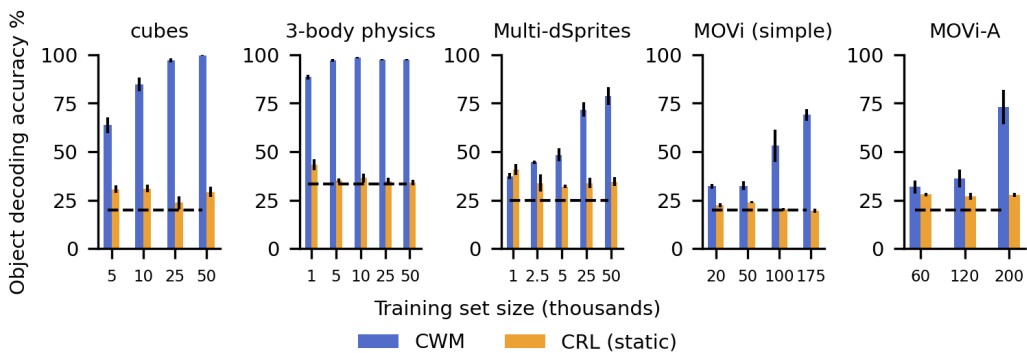

Figure 3: Object decoding accuracy as a function of training set size, for contrastive models. CWM representations of objects become more linearly separable with dataset size, despite no architectural components that encourage the formation of object-centric representations. However, contrastive learning without next step prediction (CRL) does not give rise to object-centric representations, suggesting an important role for information provided by dynamic data. Scores are averaged over five seeds (three seeds in the MOVi domains), with error bars depicting standard error of the mean.

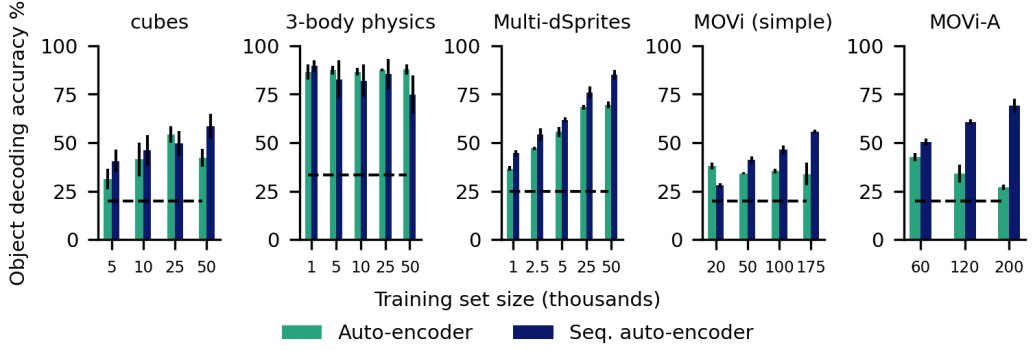

Figure 4: Object decoding accuracy as a function of training set size, for auto-encoding models. The dynamic training scheme yields a monotonic increase in object separability with training set size in four out of five datasets. Scores are averaged over five seeds (three seeds in MOVi domains), with error bars depicting standard error of the mean.

If models without object-slots can successfully generalize about object dynamics in combinatorially novel scenarios, is this because they too develop separable and compositional representations of objects? We evaluated the degree to which CWM's representations of objects were linearly separable.

Here we observe that representations of objects get more and more separable with a linear decoder as the models are provided with more training examples. In simpler domains like the `cubes` and `3-body physics`, the models attain scores close to a $100\%$ in the largest data setting (see Figure 3). In the more challenging domains like `Multi-dSprites` and the MOVi environments, where multiple objects are moving and interacting simultaneously, the decodability is lower, but substantially larger than chance at around $70\%$. For comparison, evaluating randomly initialized networks with the same metric only gives slightly better than chance object decodability scores, meaning that default representations for these models are strongly entangled (see Appendix C). Moreover, we see the same trend where larger training set sizes translate to better decodability. This suggests that, even for complex datasets with multiple interacting, realistically rendered objects, systematic and separable representations of objects can potentially emerge with scale.

To assess the importance of next-state prediction, we evaluate the object-separability of the CRL's representations. Surprisingly, the CRL attains separability scores that are close to chance, suggesting that training on dynamic object data offers valuable information for learning composable representations in the contrastive setting.

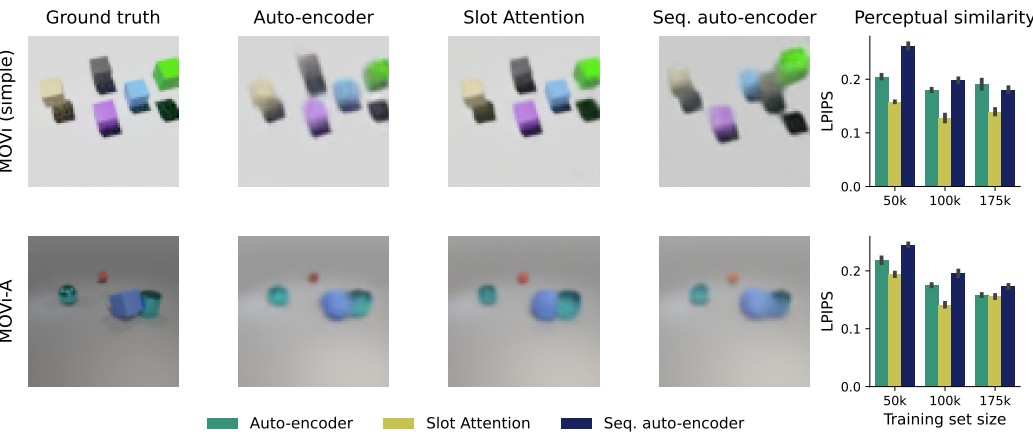

Figure 5: Reconstructions and LPIPS similarity for different models on the `MOVi (simple)` and `MOVi-A` datasets. Auto-encoding models without object slots approach or match the reconstruction ability of Slot Attention on novel object configurations in the MOVi domain.

Do these trends hold for models with non-contrastive learning objectives? We evaluated the static and sequential variants of the auto-encoding models. First, we observe a significantly stronger tendency for the *sequential* auto-encoder to develop separable object representations (see Figure 4). This also suggests that providing the models with information about object dynamics in the form of a training signal can facilitate the development of composable object representations. In fact, it is only in the `Multi-dSprites` domain that the static autoencoder shows a monotonic increase in object separability with training set size. Comparing auto-encoding objectives to the contrastive objective, we see that object separability was generally lower for auto-encoding models in the `cubes` and `3-body physics` datasets.

Next we assessed the reconstruction quality of the static and sequential auto-encoder, and compared them to Slot Attention. We used the LPIPS (Zhang et al., 2018) perceptual similarity metric to quantify reconstruction fidelity on novel object configurations in the `MOVi-A` and `MOVi (simple)` datasets. While we see that Slot Attention has a small edge on the distributed models in terms of fidelity, both the static and sequential auto-encoder approach Slot Attention with more data (see Figure 5). Lastly, the auto-encoder performs better than the sequential auto-encoder on the test set, which might be explained by it having an extra objective in the loss function.

## 5 THE BENEFITS OF PARTIALLY ENTANGLED REPRESENTATIONS

Even though unsupervised training on images of objects leads to linearly decodable representations of objects, especially in the dynamic model class, the representations of objects do not ever become

completely disjoint (see Figure 6). That is, the models rely on distributed codes in their latent spaces that often represent distinct objects using overlapping populations of neurons. Yet this does not seem to impact the models' ability to perform compositional generalization, e.g. predict dynamics and reconstruct scenes of novel compositions of objects.

To get a better qualitative understanding for the degree of object separability, we analysed the trained CWM's and their slotted counterparts' representations and the degree to which they showed systematic similarities. We obtained object representations of 300 initial frames $x_t$ and successor frames $x_{t+1}$ where only one object changed in one aspect from $t$ to $t + 1$ in the `cubes` and `Multi-dSprites` domains. We chose these domains since they had actions that accompanied the dynamics. Earlier, we used the absolute difference between these representations $|\Delta|$ to get a sense of how objects were represented. However, one could also use these absolute differences to get a sense of how *transformations* or *actions* that acted on objects were represented. For instance, pushing the red cube along the $y$-axis on the grid might cause a similar change in representation as pushing the *blue* cube along the $y$-axis, even though the same action is applied to different objects. Analogously, in the `Multi-dSprites` domain, shrinking or rotating the heart sprite could induce similar representational changes as shrinking and rotating the square sprite. We do not expect to see this for slotted models, as they are more likely to represent the properties of different objects in orthogonal sub-spaces.

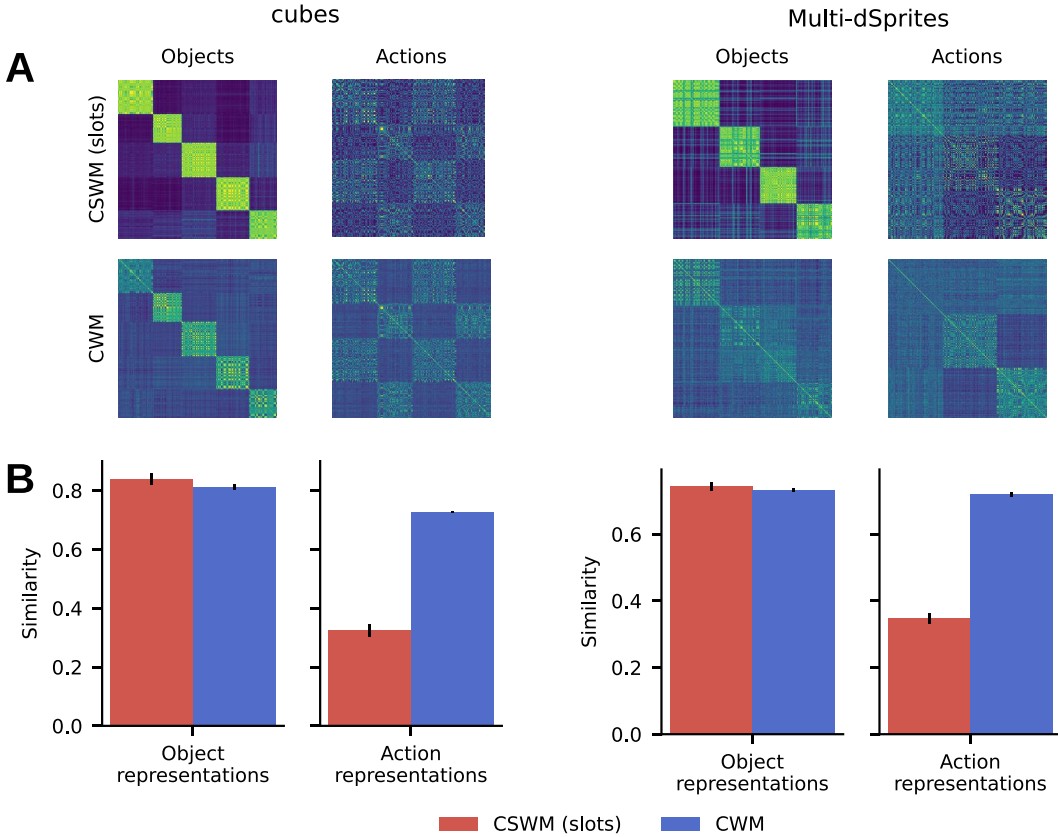

Figure 6: **A**: Representational similarity matrices showing cosine similarity of state transitions $|\Delta|$ for the CSWM and CWM on the cubes (left) and Multi-dSprites datasets (right). The cosine similarities are either ordered according to which object changed, or which action was performend on one the objects. In both cases, clusters are visible, though object clusters are more prominent in the slotted models, and action clusters more prominent in the distributed models. The cosine similarities are averaged over five seeds. **B**: CSWM intra-object similarity is significantly higher than its intra-action similarity, since object dynamics are isolated in separate subspaces. On the other hand, the CWM's intra-action similarities are much closer to the intra-object similarities, allowing for richer generalization while preserving object separability.

For both model types and datasets we obtained the absolute representational change $|\Delta|$ for all 300 frame pairs and computed pairwise similarity matrices using cosine similarity as our metric (see Figure 6A). These matrices contained information about which *transitions* were represented similarly for both distributed and slotted models. First we sorted these similarity matrices according to *which object* changed. Here we see five clear object clusters for both models in the `cubes` domain, and four clusters in the `Multi-dSprites` domain, albeit to a lesser degree for the distributed models.

Next, we sorted the similarity matrices according to *which transformation or action* was applied to the objects. While action clusters are identifiable for the CSWM, intra-action similarities were significantly lower than for the CWM (see Figure 6B). Representing object properties in a shared representational space not only allows for systematic representations of objects, but can also give rise to systematic representations of *transformations* that act on objects.

## 6 DISCUSSION

### 6.1 LIMITATIONS

Our study has focused on unsupervised representation learning models, paired with static and dynamic prediction objectives. However, the space of unsupervised learning techniques is vast. Future work should investigate object-separability in *self-supervised* representation learning settings (Grill et al., 2020; Zbontar et al., 2021; Schwarzer et al., 2020). Moreover, other model architectures like Vision Transformers (Dosovitskiy et al., 2020) are promising, as their attention patterns have been shown to match segmentation masks of natural images of objects (Caron et al., 2021).

The degree to which these properties scale to naturalistic and real-world video datasets is unclear. A natural next step is to compare larger slotted architectures, such as VideoSAUR (Zadaianchuk et al., 2024), SAVI++ Elsayed et al. (2022) and PLS (Singh et al., 2024), to distributed models on naturalistic videos. It is possible that in these complex, real world domains, relying on richer, more entangled representations can facilitate generalization. We also observed that slotted models performed better in downstream control tasks, suggesting that object-centric priors can be important in this setting. Lastly, regularization and information bottleneck methods may significantly aid in learning separable object representations as well (Alemi et al., 2016; Shamir et al., 2010).

### 6.2 CONCLUSION

We have shown that models without object slots can learn object representations that are disentangled enough to be *separable*, but entangled enough to support generalization about *transformations* of objects. Furthermore, training models to predict object dynamics significantly improved object separability. We believe our findings are important because they highlight multiple ways in which a representation can be beneficial for generalization: Slotted models can seamlessly decompose the world into its constituent objects, facilitating compositional generalization. Models with simpler, unconstrained latent spaces can decompose the world in ways that also separate objects, while allowing information about one object's dynamics and properties to permeate to others.

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

# A    Appendix: Additional Experiments

## A.1    Distributed and slotted representational alignment increases with more data

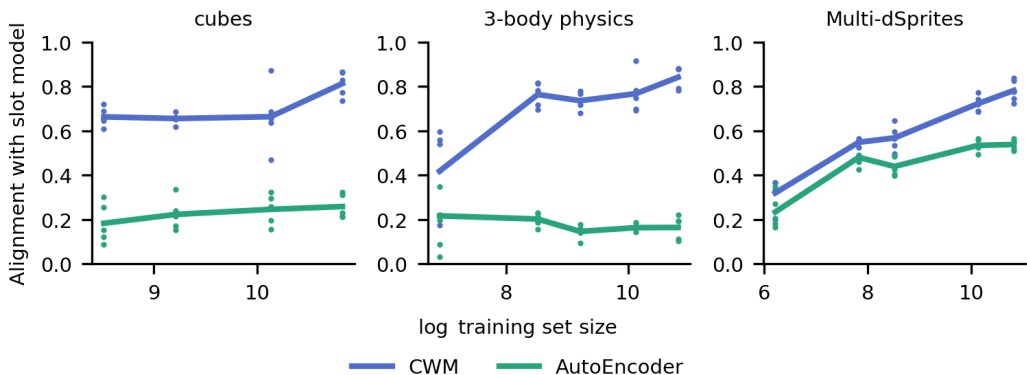

Figure 7: Alignment with slot models as determinted by representational similarity alignment (RSA) (Kriegeskorte et al., 2008; Kornblith et al., 2019). The representations of the contrastive model (CWM) become more aligned with its' slotted counterpart with more data.

It has recently been argued that deep neural network models' representations tend to grow more and more similar as training data size and model sizes increase (Huh et al., 2024). Do we see a similar convergence in latent representations of scenes composed of objects?

We extracted representations from all five seeds and all dataset sizes for CWM and the Auto-encoding models in the `cubes`, `3-body physics` and `Multi-dSprites` domains, and compared them to the corresponding representations of the object-centric CSWM models. Next we computed the Euclidean distance between all pairs of representations for all models in the three different domains, and then calculated the degree of correlation between these distance matrices of the different models. If two models' distance matrices were highly positively correlated, they perceived the same pairs of images as similar and dissimilar. In other words, their representations show a high degree of *alignment*.

In all three domains we see that the CWM's representations grow more and more aligned to those of the CSWM with more data, reaching a score of around 0.8 on average in the largest data setting (see Figure 7). This indicates that, while alignment can increase substantially with enough data, simple architectural features can leave gaps, as for instance shown in Section 5. We did not observe this trend with the auto-encoder, which showed a significantly lower level of alignment with the CSWM. This suggests that, while data can drive alignment, this has to be paired with an appropriate loss function.

## A.2    Compositional generalization

We conducted two experiments targeting compositional generalization in the MOVi environment: in the first experiment, we trained Transformer based CSWM and CWM to predict dynamics of up to four spheres, and tested on dynamics of five to eight spheres with randomly sampled colors, but identical physical attributes. We generated 3000 train videos and 600 test videos. We see that, as we increase the training set size, the disparity between the train and test accuracy diminishes, suggesting that models without slots can perform compositional generalization about scenes with more objects than in the training set (see Fig 8 Left).

In the second experiment, we trained the same models to predict dynamics of two red cubes and two green spheres, and tested on the dynamics of two green cubes and two red spheres. Again we generated 3000 train and 600 test videos. Here we see that the train-test disparity gradually

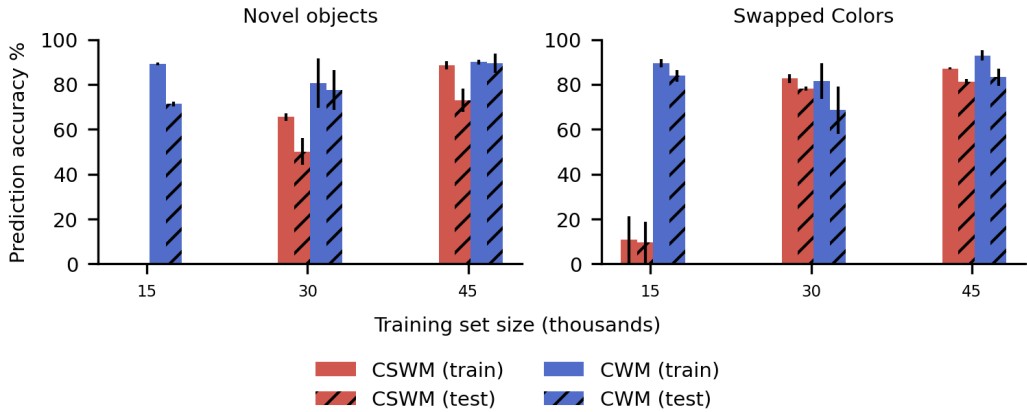

Figure 8: Both CWM and CSWM can generalize about the dynamics of i) scenes with more objects than they were trained on (Left), and ii) scenes where the objects have a novel combination of colors from the training set.

diminishes with more and more training data for both CSWM and CWM (see Fig 8 Right). CWM retains an edge in sample-efficiency in both experiments.

### A.3 DOWNSTREAM TASK PERFORMANCE

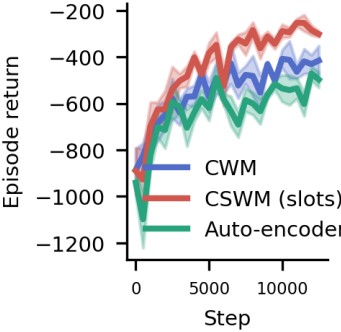

Figure 9: CSWM representations are advantageous for downstream object controllability.

We assessed the downstream task performance of our pretrained models. Here we constructed two new tasks - one control task and one prediction task. In the control task, we train a Soft Actor Critic (SAC) Haarnoja et al. (2018) agent to manipulate a randomly sampled sprite in the Multi-dSprite environment to go to a particular location on the grid. The SAC agent receives observations that are the encodings of the scene produced by one of the pretrained models. To implement the policy we use a standard MLP mapping latent representations to actions. For CSWM we concatenate object slots before passing it to the policy network since it learns aligned, temporally consistent object slots. One could replace the MLP with a Transformer or a GNN to potentially attain higher performance. We evaluate the agent using the embeddings of CSWM, CWM and the auto-encoder, and see that the agent trained to perform control using CSWM representations performs the best, with the CWM-based agent trailing closely behind (see Fig 9). This suggests that slotted representation could offer advantages in control tasks like this.

In the second downstream task we used the representations of the trained MOVi-A models to predict a novel quantity. We froze the encoders of CSWM and CWM and trained a linear classifier to predict the number of objects present in a scene. Here we see that both models can predict object cardinality

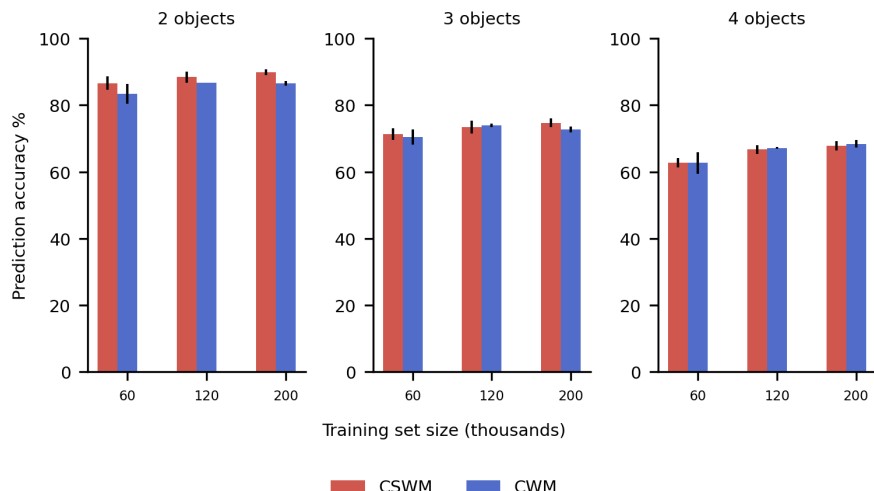

Figure 10: Object cardinality classification for CSWM and CWM.

better than chance with a simple linear classifier with only minor differences in prediction accuracy between them (see Fig 10). Moreover, prediction accuracy generally increases the more data the models were trained with in their original task.

### A.4 LONG-HORIZON PIXEL PREDICTION

We validated our forward accuracy metrics for the contrastive models by training stop-gradient decoders to reconstruct images from latent states. In the Cubes and Multi-dSprite environments we trained CNN decoders for 100 epochs to reconstruct images from the representations of CSWM and CWM, respectively. We used the converged CSWM and CWM encoders and froze their weights for the pixel reconstruction. We then evaluated the reconstruction accuracy with the LPIPS metric as the dynamics models predicted future states in an open-loop fashion. We evaluated the models for a prediction horizon of 10 steps in the future. Matching our other prediction accuracy metric, we see that CWM retains lower LPIPS for future state predictions than CSWM in both environments (see Fig

### A.5 ARE HIGH-DIMENSIONAL REPRESENTATIONS SUFFICIENT FOR OBJECT DECODABILITY?

In high enough dimensions, linear separability of a few number of classes could be trivial. This is why we install an $L_1$ norm penalty on the linear classifier weights when we perform the linear separability analysis. We further tested whether models with high-dimensional latent spaces trivially

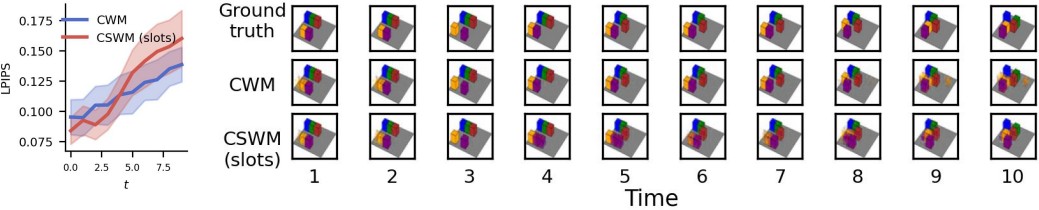

Figure 11: LPIPS and future frame reconstructions using a stop-gradient decoder on pretrained CWM and CSWM models in the cubes environment. CWM performs favorably. Shaded region represents standard errors of the mean.

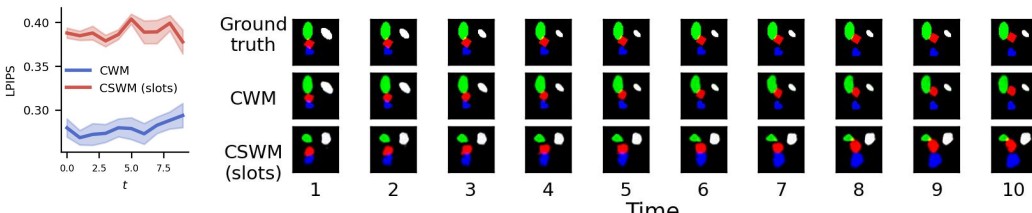

Figure 12: LPIPS and future frame reconstructions using a stop-gradient decoder on pretrained CWM and CSWM models in the Multi-dSprite environment. CWM performs favorably. Shaded region represents standard errors of the mean.

attained high separability scores. We randomly initialized image encoders with 50, 100, 500, 1000, and 2000 latent dimensions, respectively. Without training, these models perform only slightly better than chance levels, whereas a trained CWM model with 50 latent dimensions is close to ceiling performance on our separability metric (see Fig 13a).

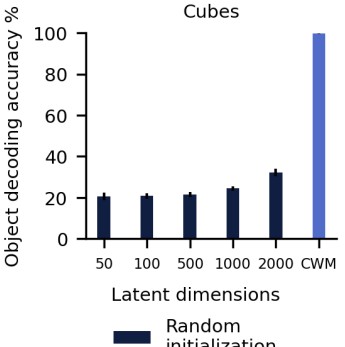

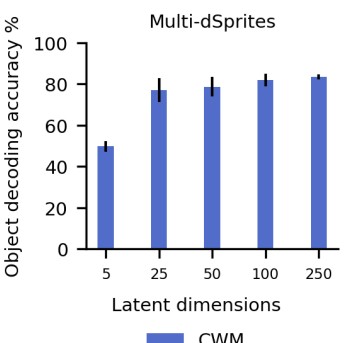

(a) High-dimensional feature spaces alone are not sufficient for object separability when decoding using an $L_1$ norm penalty on decoder weights.

(b) Object decodability saturates with latent dimension size. Too few latent dimensions leads to more entanglement.

Next we trained CWM with various latent representation sizes (5, 25, 50, 100 and 250, respectively) in the Multi-dSprite environment. While too low latent dimension sizes yielded worse accuracy, once the representational space becomes big enough (around 50), accuracy saturates, suggesting that the encoder needs to be of suitable capacity to represent objects in a disentangled way (see Fig. 13b).

## A.6 COMBINED RESULTS

For completeness we compile the results presented in the main text together here, showing a prediction accuracy comparison for all dynamics prediction models in Fig. 14, and a object separability comparison for all models in Fig. 15:

## A.7 RSA

Since CWM outperforms CSWM in the dynamic prediction tasks, we performed RSAs, comparing CWM to CSWM, as well as the static and dynamic auto-encoder models. Although the dynamic auto-encoder develops representations with a similar degree of object-separability to CWM, its representations are on average less aligned than the CSWM. This suggests that model representation may still differ in important ways despite representing objects in separable subspaces (see Fig 16).

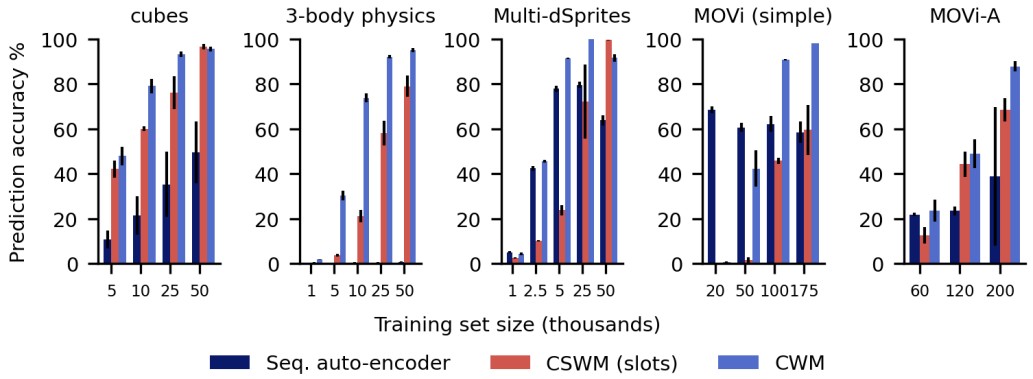

Figure 14: Prediction accuracy, including the sequential auto-encoder.

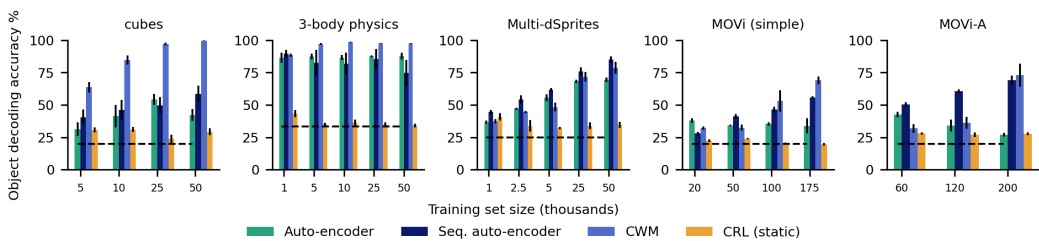

Figure 15: All object separation scores.

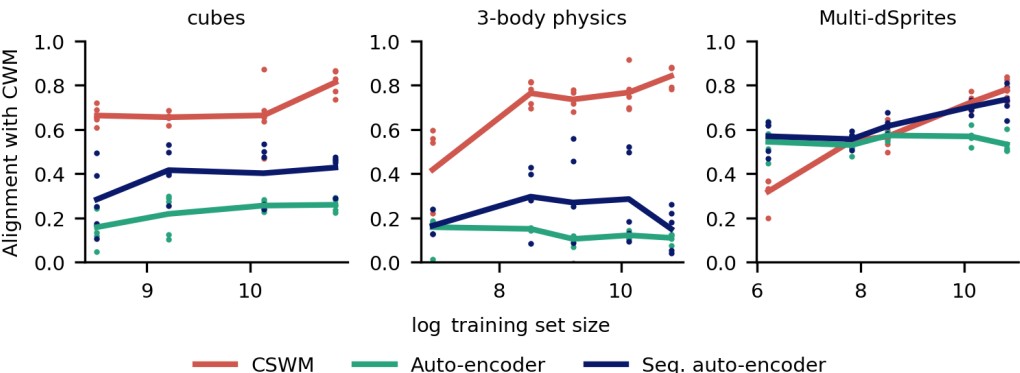

Figure 16: RSA scores between CWM and the auto-encoding models is lower, suggesting that loss function plays an important part in shaping object representations, beyond disentangling them.

## A.8 SSIM AND MSE

## B APPENDIX: ARCHITECTURE AND HYPERPARAMETERS

### B.1 CONVOLUTIONAL NEURAL NETWORKS

For our image encoders we rely on a standard CNN architecture used in other works such as Yarats et al. (2021b;a). For the 3-body physics dataset we stacked two frames, as in Kipf et al. (2019) to provide information about directional velocity.

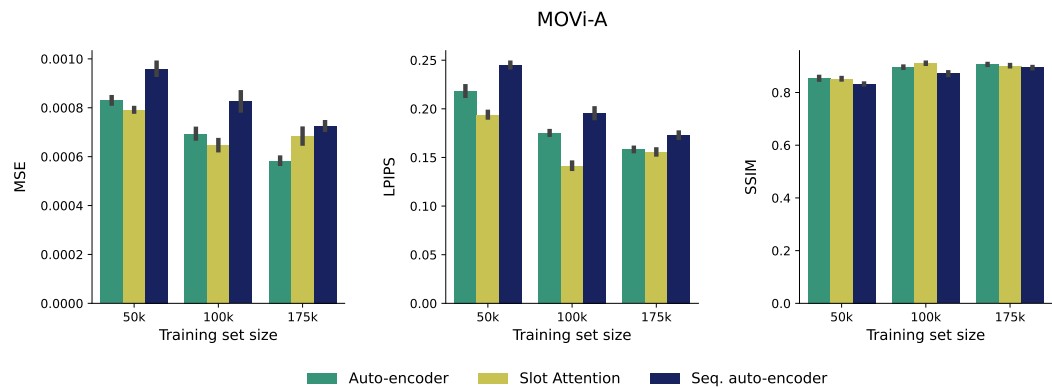

Figure 17: SSIM, LPIPS and MSE for the MOVi-a dataset.

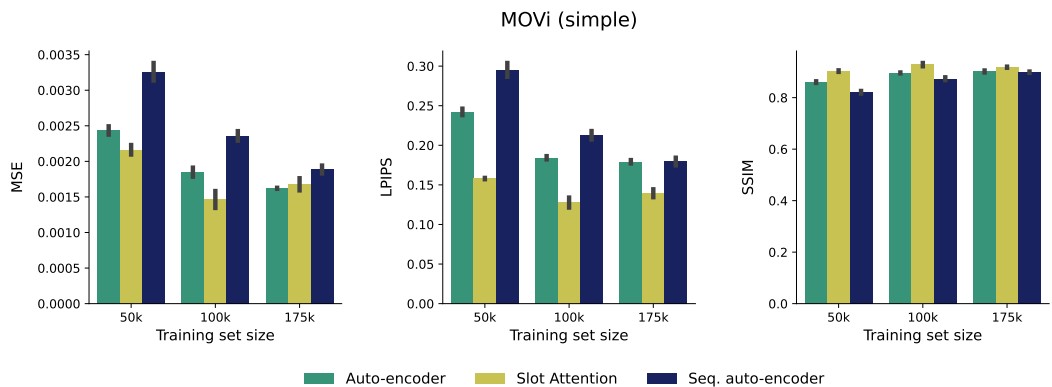

Figure 18: SSIM, LPIPS and MSE for the MOVi (simple) dataset.

```python
import torch
from torch import nn
encoder = nn.Sequential(
            nn.Conv2d(num_channels, 32, 3, stride=2),
            nn.ReLU(),

            nn.Conv2d(32, 32, 3, stride=1),
            nn.ReLU(),

            nn.Conv2d(32, 32, 3, stride=1),
            nn.ReLU(),

            nn.Conv2d(32, 32, 3, stride=1),
            nn.ReLU()
            )
```

This network is followed by an MLP with 2 hidden layers and 512 hidden units. All models used ReLU (Nair & Hinton, 2010) activation functions and were optimized using Adam Kingma & Ba (2014). The CSWM and Slot Attention models were trained using the hyperparameters and encoder architectures provided in Kipf et al. (2019) and Locatello et al. (2020b), respectively. Number of model parameters are provided below.

Table 1: Trainable parameters for all models in the representative Cubes environment.

| Model | Trainable parameters |
|---|---|
| CSWM | 2.5M |
| CWM | 5.9M |
| Context length | 8M |
| Sequential auto-encoder | 8.1M |

## B.2    TRANSFORMER

We use the causal Transformer architecture of GPT-2 (Radford et al., 2019), building on the implementation in the `nanoGPT` repository[1]. In the slotted dynamics model, the transformer applied attention over the sequence of slots per time-step. Assuming $K$ slots and $T$ time-steps, the transformer applied attention over a sequence of $K \times T$ data-points. In `MOVi (simple)` we trained CSWM with 6 object slots, allowing each slot to model one object plus a background slot. In `MOVi-A` we trained CSWM with 11 object slots, allowing for 10 separate object representations plus the background. The Slot Attention model was trained with the same number of slots as CSWM. In total, the transformer models had a total on 6.8 million trainable parameters. The transformers were trained with the following hyperparameters:

Table 2: Transformer hyperparameters.

| Hyperparameter | Value |
|---|---|
| MLP Hidden units | 2048 |
| Transformer blocks | 2 |
| Context length | 4 |
| Heads | 8 |

## B.3    HYPERPARAMETERS

Below are specific hyperparameters for the different distributed model classes.

Table 3: Contrastive model hyperparameters.

| Hyperparameter | Value |
|---|---|
| Hidden units | 512 |
| Batch size | 512 (1024 for MOVi) |
| MLP hidden layers | 2 |
| Latent dimensions $|z_t|$ | 50 (500 for MOVi) |
| Margin $\lambda$ | 1 (100 for MOVi) |
| Learning rate | 0.001 (0.0004 for MOVi) |

For auto-encoding models we use the transpose of the encoder networks presented above. These models are trained with the following hyperparameters:

Table 4: Auto-encoder hyperparameters.

| Hyperparameter | Value |
|---|---|
| Hidden units | 512 |
| Batch size | 512 (64 for MOVi) |
| MLP hidden layers | 2 |
| Latent dimensions $|z_t|$ | 50 (500 for MOVi) |
| Learning rate | 0.001 (0.0004 for MOVi) |

---

[1]See https://github.com/karpathy/nanoGPT

Table 5: Sequential auto-encoder hyperparameters.

| Hyperparameter | Value |
|---|---|
| Hidden units | 512 |
| Batch size | 512 (124 for MOVi) |
| MLP hidden layers | 2 |
| Latent dimensions $\|z_t\|$ | 50 (500 for MOVi) |
| Learning rate | 0.001 (0.0004 for MOVi) |

# C  OBJECT DECODABILITY BASELINE

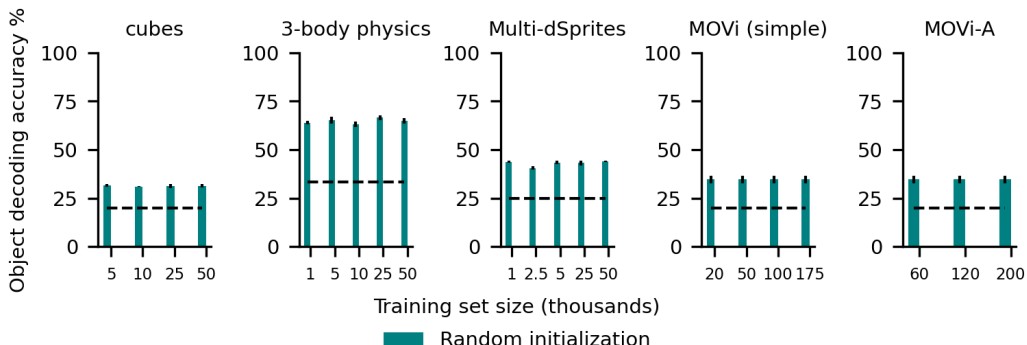

Figure 19: When initializing the image encoders randomly, the linear decodability of objects is higher than chance, but substantially lower than the level of trained models.

## C.1  SLOTTED REPRESENTATIONS ARE RECOVERABLE FROM DISTRIBUTED REPRESENTATIONS

We assessed the degree to which CWMs representations could be mapped to discrete object slots in the cubes dataset. We trained a slot decoder network with the same architecture as the Slot Attention model to reconstruct images from the representations of CWM. Assuming $K$ object slots, the decoder was trained to reconstruct the original frame as an additive composition of $K$ individual objects. Crucially, we froze CWM's encoder, meaning that the decoder could only use information learned through the contrastive dynamic training to recompose the scene. We see that the slot decoder can not only learn to reconstruct the scenes with high fidelity, but also reconstruct the scenes by individually reconstructing the cubes and composing them.

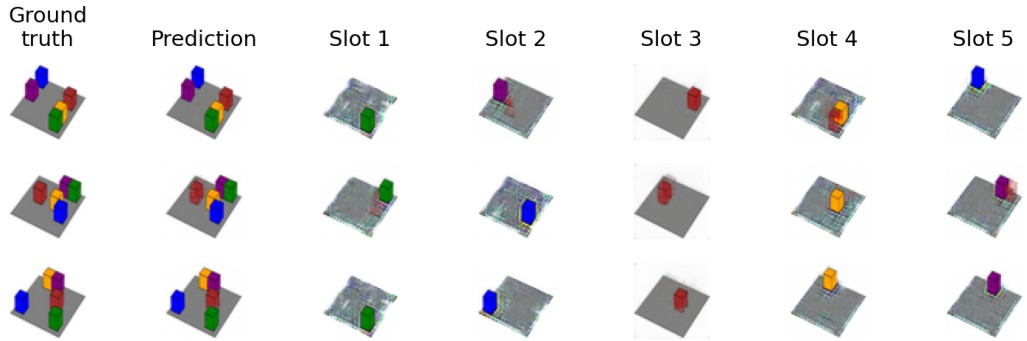

Figure 20: Training a slot-decoder to reconstruct scenes from CWM's learned representations leads to scene re-compositions that track the original ground truth objects.

