# OpenReview forum: "Next state prediction gives rise to entangled, yet compositional representations of objects"
_ICLR.cc/2025/Conference — Submitted to ICLR 2025_

### Official Review · Reviewer_v3zg · 2024-10-27

**Soundness:** 1
**Presentation:** 3
**Contribution:** 2
**Rating:** 3
**Confidence:** 5

**Summary:**

This paper explores how distributed representation models can develop compositional, linearly separable object representations without object-centric architectures/inductive biases. Through next-state prediction, the authors claim that these models match or exceed the performance of slot-based models in predicting object dynamics, even without dedicated object slots. The study finds that partially overlapping neural codes in distributed models enable effective generalization and object separability, making them a viable alternative for tasks involving dynamic object interactions.

**Strengths:**

The paper’s strengths lie in its idea of challenging the necessity of object slots, trying to show distributed models can effectively generalize with next-state prediction alone. It provides robust evidence across diverse datasets and offers new insights into how partial neural overlap supports transformation generalization, broadening representation learning approaches.

**Weaknesses:**

I want to remind the authors that, it is unfair to compare your model with small models designed for a special dataset, like CSWM, on new datasets such as multi-dsprites.

Thus, The paper exhibits several weaknesses in supporting its claims:

1. The claims about distributed models achieving compositional representations are not robustly supported by the experiments. The CSWM baseline, a model specifically optimized for the datasets used in their studies, underperforms on the new datasets used in this paper. This performance drop is predictable and weakens the validity of the comparisons.

2.  The study relies on Slot Attention, which is designed for static images, as a baseline in dynamic scenes. For a fair comparison in dynamic settings, it should have considered more relevant recent work, such as Parallelized Spatiotemporal Binding, which better aligns with the dynamical nature of the datasets.

3. The claim that next-token or next-state prediction without specifically designed inductive biases can lead to a (somewhat) disentangled representation is not new. This phenomenon has been observed and documented in previous works, making the findings here less innovative.

Overall, the paper lacks sufficient experimental depth, relevant benchmarks for dynamic contexts, and originality in its claims, suggesting it may not yet be ready for acceptance in its current form.

**Questions:**

In Eq.6 , the t should be replaced by t+1?

---

> ### Author Response · Authors · 2024-11-21
>
> We thank the reviewer for engaging with our work. However, we were quite puzzled about this review. Although the reviewer first states that our paper “provides robust evidence across diverse datasets”, they later write that “The claims about distributed models achieving compositional representations are not robustly supported by the experiments.“. The reviewer also claimed that “it is unfair to compare your model with small models designed for special dataset, like CSWM, on new datasets such as multi-dsprite”. We firmly disagree with this statement, as the Multi-dSprite environment we construct is very similar in character and complexity to the Cubes and 3-body Physics datasets CSWM was trained on in the original paper. Moreover, we even show in Figure 2 that CSWM can get close to a 100% test accuracy when trained on 50k transitions, suggesting that it is well-calibrated to deal with this dataset. Finally, the CWM model we compare it to only differs from the CSWM model in that it uses an MLP for dynamics prediction instead of a GNN, and uses a generic CNN encoder instead of one mapping images to slots. We therefore do not believe that a rating of 1 with confidence 5 is justified. We go through the reviewer’s comments in more detail below.
>
> > The CSWM baseline, a model specifically optimized for the datasets used in their studies, underperforms on the new datasets used in this paper.
>
> We do not believe that CSWM underperforms. In the dSprite environment it eventually attains close to 100% test accuracy when given enough data. In other environments, like 3-body physics, our results for CSWM match those reported in the original paper - still CWM outperforms CSWM, despite not having slots.
>
> >The study relies on Slot Attention, which is designed for static images, as a baseline in dynamic scenes.
>
> We want to clarify that we train the Slot Attention model as a static auto-encoder, reconstructing static scenes from two MOVi-based datasets. We only evaluate it on its ability to reconstruct static scenes, not to perform dynamic prediction.
>
> >The claim that next-token or next-state prediction without specifically designed inductive biases can lead to a (somewhat) disentangled representation is not new. This phenomenon has been observed and documented in previous works, making the findings here less innovative.
>
> We would like to encourage the reviewer to reference the papers they had in mind so we can assess the novelty of our findings. We are unaware of any previous works showing that next-state prediction can produce separable representations of objects. We also note that the novelty of our work was highlighted by all other reviewers.
>
> >In Eq.6 , the t should be replaced by t+1?
>
> This equation refers to the static Auto-Encoder, which does not perform next-state prediction. It is only trained to reconstruct static images, as is the Slot Attention model.

---

> ### Comment · Reviewer_v3zg · 2024-11-22
>
> I want to clarify that I agree that the paper provides robust evidence in terms of numbers, but I have very strong doubts towards the conclusion due to  the baselines.
>
> If you read the CSWM paper, you will find that their main contribution is about the training loss --- the constrasive loss, not the neural architecture. In fact, their neural architecture, while seems to be "slot based", is in fact very adhoc--- they just used the different channels of a convolutional layer as "slots". It is not surprising that their reconstruction does not perform very well.
>
> The main selling point of the slot attention paper is about the disentangled representations, not about accurately predicting object dynamics. Which is quite understandable to me why they underperforms in the paper's experiments.
>
> I still think this paper's baseline are too weak. Both CSWM and slot attention, at least in their original form, are TOY models designed for their specific task, they are not general purpose models like transformers. These models are NOT meant to be generalizable to other tasks and datasets. It makes no sense to me to compare with them, except you use the exact same dataset and criterion used in their paper.
>
>  I dont think you can draw any conclusions just comparing with them but not with recent scaling up versions like SAVI++ and PSL.
>
> I maybe too negative about the paper in the beginning, I think the claim that "next token predict can result in linearly separable representations" is well supported.  But this is not that new. Predicting the future is a good objective for learning video representations is a VERY well known fact among deep learning community. For example, the 2019 paper Video Representation Learning by Dense Predictive Coding used similar loss for video representation learning. I think the only novelty is to pose the problem specific to object centric case.  Surprising? Not for me.
>
> But the claim that "the proposed model can match or even better than slot based algorithms in downstream tasks" is not  well supported. The reason is that the baselines are too weak.
>
> I am raising the score to 3, but I still dont think this paper should be published in the current form.

---

> > ### Author Response · Authors · 2024-11-22
> >
> > Thanks for the swift response and for your clarifications. We also appreciate that the reviewer updated the score to 3, and we would like to remind the reviewer to update the score in the console as well. We would like to re-emphasize a few points:
> >
> > >If you read the CSWM paper, you will find that their main contribution is about the training loss --- the constrasive loss, not the neural architecture. In fact, their neural architecture, while seems to be "slot based", is in fact very adhoc--- they just used the different channels of a convolutional layer as "slots". It is not surprising that their reconstruction does not perform very well.
> >
> > We evaluate CSWM using exactly the same metrics and methodology used in the original paper (latent prediction accuracy), not on pixel reconstructions. We compare CSWM agains CWM on two datasets from the original paper (Cubes and 3-body physics), and in both of these CWM attains higher accuracy with less data. This same trend holds for the other datasets too.
> >
> > >The main selling point of the slot attention paper is about the disentangled representations, not about accurately predicting object dynamics. Which is quite understandable to me why they underperforms in the paper's experiments.
> >
> > We do not evaluate Slot Attention on dynamic prediction, we simply evaluate how well it reconstructs novel object configurations.
> >
> > > Both CSWM and slot attention, at least in their original form, are TOY models designed for their specific task, they are not general purpose models like transformers.
> >
> > We would like to clarify that in the more challenging MOVi environment, we do not compare CSWM against a Transformer - we pair it with a Tranformer to predict object dynamics. Pairing slotted encoders with Transformers for dynamic prediction is a common modeling technique, used in Slotformer [1].
> >
> > >I dont think you can draw any conclusions just comparing with them but not with recent scaling up versions like SAVI++ and PSL.
> >
> > Thanks for suggesting this. SAVI++ uses depth prediction as an auxiliary objective to learn robust representations of objects. In this paper we studied the formation of object representations in a completely unsupervised setup. Other models like PSL, which was published only a few months before we submitted our paper, are interesting to evaluate too, which we are happy to do in the future. We now reference these works in our Discussion:
> >
> > **”A natural next step is to compare larger slotted architectures, such as VideoSAUR (Zadaianchuk
> > et al., 2024), SAVI++ Elsayed et al. (2022) and PLS (Singh et al., 2024), to distributed models
> > on naturalistic videos.”**
> >
> >
> > > Predicting the future is a good objective for learning video representations is a VERY well known fact among deep learning community.
> >
> > We agree that there are previous works showing that predicting the future is a good objective for representation learning. However, we show something more specific than this in our paper - we show that linearly separable representations of objects emerge with future prediction. This is surprising because many papers argue that slot-based architectures are necessary for learning compositional representations of objects. We challenge this claim and show with additional generalization results that non-slotted models can generalize compositionally (see Figure 8, page 15).
> >
> > We thank the reviewer for the discussion and hope they may still reconsider our paper.
> >
> > [1] Wu, Ziyi, et al. "Slotformer: Unsupervised visual dynamics simulation with object-centric models." arXiv preprint arXiv:2210.05861 (2022).

---

### Official Review · Reviewer_ZxZn · 2024-10-31

**Soundness:** 2
**Presentation:** 3
**Contribution:** 3
**Rating:** 6
**Confidence:** 3

**Summary:**

This paper investigates the quality and content of the latent representation of different models trained using unsupervised learning on dynamic scenes composed of multiple objects. The authors compare slot-based models against distributed models and investigate their ability to encode object identity and object dynamics (movement in the scene). They find that distributed models have a better ability to encode identity *and* movement than slot-based models.
They also show that the models trained with more specific losses, such as contrastive loss and next-state prediction, help to improve the quality of these representations through linear readout.

**Strengths:**

- The paper is overall clearly written and well presented.

- The question tackled by the authors is very interesting and has not been extensively investigated in the past.

- The method proposed to investigate the content of the latent representation by systematically controlling which object moves from one frame to the other and the nature of the movement is a very elegant way to test the information encoded in the latent representation.

- The authors propose various ways to evaluate the content of the representations with different metrics, simple yet elegant and informative.

- The number of datasets and their variety in content and difficulty solidifies the authors' conclusions.

**Weaknesses:**

The results part should be improved. Each experiment compares only two models (either in terms of objective or model type -- slot-based vs. distributed), but we miss a big picture comparing the baseline (the CSWM) with all the models with their characteristics (contrastive, next-state prediction, ...).
It would be easier to read and draw conclusions if all the models appeared on all the bar charts, clearly highlighting the benefit of each objective depending on the task, assuming that all the models are comparable in terms of number of parameters and architectural design.
Maybe this could be done for one of the datasets, and the other results can appear in the appendix.
It would also be informative to add in the Appendix the complete architecture of all the models, along with their number of parameters.
I will be willing to increase my rating if this part is improved with a figure showcasing the totality of the results, for all models on all metrics, backing the claims of the authors.

**Questions:**

- For the dynamic loss (L_{AE-dynamic}), have the authors tried each term separately in addition to both combined? Does it change the conclusions for dynamic AE models?

- Instead of plotting the different metrics with respect to the number of training data, it would be interesting to do the same for various sizes of latent representation. Augmenting the capacity of this latent space will affect the information contained, making it more or less distributed -- and potentially showing different scores on the metrics. Have the authors tried this?

- Even though Fig. 6 seems to show that object identity is encoded in distributed models, is it possible to perform a linear to decode only object identity for the first or last frame of the videos? Slot models should be able to decode all the objects, but maybe the distributed models will only be able to decode the ones that are moving?

- In Fig. 2, the accuracy of both models increases with more data, but the gap seems to decrease. Is there a point with even more data where the CSWM is as good as the CWM?


- Fig. 7, why choose the CSWM as the baseline for the RSA if it's not the best model? It would be interesting to perform this analysis on all the models with respect to the CWM which seems to perform best on all the metrics, to see which component plays the biggest role in explaining the quality of the representations.

---

> ### Author Response · Authors · 2024-11-21
>
> We thank the reviewer for their thoughtful and encouraging review of our paper. We are glad the reviewer found our paper well written and interesting. The reviewer also provided some important suggestions for improving the presentation of our results. As a consequence we have added a figure summarizing all dynamics models’ scores for our prediction accuracy metric for all datasets. We also added a figure summarizing all the reported linear separability scores into one figure for all datasets. We address the reviewer’s points in more detail below.
>
> >For the dynamic loss (L_{AE-dynamic}), have the authors tried each term separately in addition to both combined?
>
> We did not experiment with ablating the latent consistency loss, as this loss term has been shown to be important in other works [1, 2]. Ablating the pixel reconstruction loss would lead to representational collapse, as the encoder can minimize the other loss term by mapping all observations to a constant vector [3].
>
>
> > Instead of plotting the different metrics with respect to the number of training data, it would be interesting to do the same for various sizes of latent representation. Augmenting the capacity of this latent space will affect the information contained, making it more or less distributed -- and potentially showing different scores on the metrics. Have the authors tried this?
>
> Thanks for this interesting suggestion. We trained CWM with various latent representation sizes (5, 25, 50, 100 and 250, respectively) in the Multi-dSprite environment. While too low latent dimension sizes yielded worse accuracy, once the representational space becomes big enough (~50), accuracy saturates, suggesting that the encoder needs to be of suitable capacity to represent objects in a disentangled way (see Figure 13B, page 17).
>
> >the accuracy of both models increases with more data, but the gap seems to decrease. Is there a point with even more data where the CSWM is as good as the CWM?
>
> Indeed, CSWM catches up with the CWM in the Cubes and Multi-dSprite environments when trained on 50k transitions. Adding more data in the other environments is therefore likely to continue to increase CSWM performance.
>
>
> > why choose the CSWM as the baseline for the RSA if it's not the best model? It would be interesting to perform this analysis on all the models with respect to the CWM which seems to perform best on all the metrics, to see which component plays the biggest role in explaining the quality of the representations.
>
> Thanks for this suggestion. We added a figure where we show how well CSWM, as well as the static and dynamic auto-encoder models align with CWM. Although the dynamic auto-encoder develops representations with a similar degree of object-separability to CWM, its representations are on average less aligned than the CSWM. This suggests that model representation may still differ in important ways despite representing objects in separable subspaces. See Figure 16, page 18.
>
> > It would also be informative to add in the Appendix the complete architecture of all the models, along with their number of parameters.
>
> We thank the reviewer for the suggestion, we report the architecture, hyperparameters and parameter counts for the models in Appendix B, page 19.
>
> [1]Watter, Manuel, et al. "Embed to control: A locally linear latent dynamics model for control from raw images." Advances in neural information processing systems 28 (2015).
>
> [2]Hafner, Danijar, et al. "Dream to control: Learning behaviors by latent imagination." arXiv preprint arXiv:1912.01603 (2019).
>
> [3]Schwarzer, Max, et al. "Data-efficient reinforcement learning with self-predictive representations." arXiv preprint arXiv:2007.05929 (2020).

---

> > ### Comment · Reviewer_ZxZn · 2024-11-25
> >
> > I first thank the authors for replying to the points raised in my review.
> >
> > I acknowledge that Figures 14 and 15 partly answer the weakness raised in the review. However, it has not fully resolved my concerns, see the details below:
> > - Why isn't the auto-encoder present in Fig. 14? If the authors use four models to present the results, the four models should always appear in all the figures. Additionally, I think these figures should appear in the main text instead of the Appendix.
> > - The additional experiment with different latent dimensions is interesting and would be even more informative if performed on all the baselines.
> > - Most importantly, these figures were supposed to help compare the models to clarify the conclusions of the authors, "assuming that all the models are comparable in terms of number of parameters and architectural design" (cited from my review). However, it turns out that the models are not comparable in terms of the number of parameters, because the CWM has more than twice the number of parameters as the CSWM and far fewer parameters than the sequential auto-encoder. Consequently, the comparisons are very unfair, unless it is proven that the number does not affect the results. For this reason, I would urge the authors to run control experiments with baselines comparable in terms of the number of parameters.
> >
> > For these reasons, I will not change my score until these points are resolved.

---

> > > ### Author Response · Authors · 2024-12-02
> > >
> > > Thanks for the additional suggestions and clarifications.
> > >
> > > > Why isn't the auto-encoder present in Fig. 14?
> > >
> > > The auto-encoder is only trained to reconstruct the current state, not to predict the next state. Since the auto-encoder has no means of predicting the next state, we cannot compare it to the dynamics models in terms of its accuracy in predicting future latent states in the environments. Figure 14 compares all models that are comparable on this metric.
> > >
> > > > the CWM has more than twice the number of parameters as the CSWM
> > >
> > > This is a good point. To address this we reduced the MLP width from $512$ to $128$ hidden dimensions in the dSprite environment. In this setting CWM has $2.6$M parameters, which is approximately what CSWM has. Training the smaller CWM model on 50k transitions in the Multi-dSprite dataset gives close to ceiling prediction accuracy $accuracy = 95.4$, suggesting that our effects are not do to larger model sizes. We would include full results in a camera-ready version.
> > >
> > > Thanks again for the constructive feedback.

---

### Official Review · Reviewer_xykj · 2024-10-31

**Soundness:** 2
**Presentation:** 3
**Contribution:** 3
**Rating:** 5
**Confidence:** 4

**Summary:**

This work performs a study comparing the learned representation spaces of unsupervised object-centric slot-based models and their unstructured counterparts trained on various datasets of dynamically interacting objects. The study focuses on the disentanglement of these representations in terms of objects and their capacity to capture underlying transformations that act on objects, regardless of their identity. The experiments include models trained on both reconstruction-based and contrastive objectives and test the learned representation’s efficacy in image reconstruction as well as multi-step dynamics prediction. Object separability is measured using a novel metric based on the ability of a linear classifier to recognize which object changed between two consecutive frame representations. Based on their experiments and analysis, the authors suggest that explicit object-level factorization is not necessary for learning these tasks and achieving compositional generalization, more so when the training objective includes next-state prediction. This is explained by the fact that the learned representations, although not object-centric, converge to being partially disentangled in terms of objects (as quantified by the object separability metric) while maintaining a level of entanglement that facilitates sharing of information about transformations that act on objects.

**Strengths:**

Summarized points:
- Well written paper
- Perform an interesting and seemingly novel study
- Revisits and challenges an important basic assumption that incorporating explicit object-centric structure in representation-learning models is beneficial/necessary for representing object-centric scenes/dynamics

**Weaknesses:**

Summarized points:
- Some conclusions are not convincingly backed-up by experiments
- There is room for more systematic fine-grained evaluation of performance with respect to different types of compositional generalization
- The implications of this study to downstream task performance, a major use-case/motivation for unsupervised representation-learning, are not assessed in experiments and mostly not discussed

**Related Work**

The overview of object-centric representations is focused entirely on slot-based models. Although these might be the most dominant in the literature, I believe this section can benefit from including additional types such as patch-based (e.g. [SPACE](https://arxiv.org/abs/2001.02407), [MarioNette](https://arxiv.org/abs/2104.14553)) and keypoint-based (e.g. [Jakab et al.](https://arxiv.org/abs/1806.07823), [DLP](https://arxiv.org/abs/2205.15821)).

**Experiments - Model Forward Prediction Ability**

*Prediction Accuracy*: The evaluation metric described in the second paragraph of the Experiment Section is not clear to me. What does it mean for the prediction to be “closest to the last frame”, closest compared to what?

In the case of the object-centric dynamics model, how do you compute the distance between two latent representations that are not from the same trajectory (e.g. contrastive examples or encoded ground-truth future states)? The latents are sets of vectors and thus lack ordering such that a simple L2 loss would not necessarily compute distances between aligned slots.

I find it surprising that CWM outperforms CSWM in object dynamics prediction. The original CSWM paper seems to have results on the cubes dataset that contradict your results (see Table 1 -> 3D Blocks -> 10 Steps -> CSWM vs. -factored states). Do you have an explanation for this discrepancy? Is there an essential difference in the setting or metrics they used compared to yours?

Another question that arises here is why did you choose to compare the prediction abilities in latent space? The latent spaces are not as comparable as reconstructions of images in my opinion since they possess different structure. I suggest training a “stop-gradient decoder” on reconstructing the contrastive latents so you can obtain image reconstructions without affecting the contrastive model’s optimization process. You can then use perceptual metrics to quantify prediction performance (e.g. LPIPS, which you used in the autoencoder study).

Why do the authors not study object dynamics prediction with the autoencoder models in addition to the contrastive models in section 4.1? This study could potentially strengthen your conclusions. In any case, I believe it should be part of your experiments.

**Experiments - Reconstruction Quality**

Figure 5 presents similarity metrics using LPIPS. For better qualitative assessment I would suggest adding additional metrics such as pixelwise MSE, SSIM, FID.

“the auto-encoder performs better than the sequential auto-encoder on the test set, despite attaining substantially lower object separability scores” - Is it possible that this is due to the fact that the auto-encoder was trained on that task precisely while the sequential auto-encoder was trained on next-frame prediction? The distribution of latent representations produced by the latent dynamics model is most likely different from latent representations directly encoded from images.

**Experiments - Compositional Generalization**

Can the authors clarify why high accuracy in the dynamics prediction tasks suggests compositional generalization capabilities? Is there a systematic separation between the train and test data in terms of compositions of objects and/or their properties such that performing well on the test data would necessarily require compositional generalization?
Put more simply: can you say for certain or with high probability that the test data does in fact contain novel constellations and combinations of objects?
If so, I request that the authors add details about the train-test split with respect to the relevant factors of variation in each dataset.

The above questions are also relevant to the reconstruction experiments.

**Discussion**

As I see it, the two major claims about non-object-centric unsupervised representation learning models are:
- They can learn representations with some form of disentanglement between objects (as quantified by the proposed separability score)
- The learned representation space captures some notion of underlying transformations that act on objects, regardless of their identity

I believe the results that show that the object-separability score does not translate to improved task performance when comparing the static autoencoder with the dynamic one is indicative of a broader question that should be asked, discussed and maybe answered in the context of this work: *What are the implications of the two major findings in the paper?*

*I find this question not sufficiently answered*. I suggest looking into two main aspects:

1. *Downstream Task Performance*: Questions that should be answered in my opinion:
- Are the two qualities of the learned representation indicative of downstream task performance?
- Are they efficiently leverageable by downstream models?

I would argue that in this study, downstream task performance was not assessed. Both dynamics prediction and image reconstruction are exactly what the respective models were trained on.
Downstream tasks in the actionable environments could be sequential decision-making while on the uncontrollable dynamics, one could infer underlying properties of the scene or trajectory such as the number of objects that remain in the scene by the end of the trajectory in the MOVi-A dataset.

2. *Compositional Generalization*: the models’ ability to generalize its representations to scenes with novel compositions of objects as well as novel compositions of objects and the transformations that are applied to them.

Here I argue that this type of generalization was not systematically assessed.

In order to do so, I suggest first defining the factors of variation of interest.
Some examples:
- Number of objects in the scene
- Combinations of individual object static properties such as color, shape, size, mass, friction (most relevant to the MOVi-A dataset)
- Dynamic transformations in static object properties (most relevant to Multi-dSprites)

Based on these factors one should create a train-test split such that performance of the same task on the test data would be indicative of the model’s representation’s capacity to facilitate compositional generalization of that type.
Examples of train-test splits:
- Train includes up to x objects -> Test includes between x and x+y objects
- Train includes only red cylinders and blue cubes -> Test includes blue cylinders and red cubes
- Train includes sequences where all but circular objects dynamically change scale -> Test includes sequences where circular objects do change scale

**Questions:**

For major questions and requests, see weaknesses.

Additional minor questions/requests:
- What are the actions that can be taken in the cubes and Multi-dSprites environments?
- Figure 6A: could the authors add labels of the objects/actions to the similarity matrices? It would be easier to interpret them knowing this information.

---

> ### Author Response · Authors · 2024-11-21
>
> We would like to thank the reviewer for their thorough review of our paper. We appreciate that the reviewer found our paper well written and our findings novel and interesting. At the same time the reviewer raised a number of important points. These pertained to i) compositional generalization, ii) downstream tasks and iii) validating forward performance.
>
> First, we conducted two new experiments targeting compositional generalization in the MOVi environment (see Figure 8, page 15): in the first experiment, we trained Transformer based CSWM and CWM to predict dynamics of up to 4 objects, and tested on dynamics of 5 to 8 objects. In the second experiment, we trained the same models to predict dynamics of two red cubes and two green spheres, and tested on the dynamics of two green cubes and two red spheres (color flip). In both experiments we see that there’s a train-test disparity that gradually diminishes with more and more training data for both CSWM and CWM. CWM retains an edge in sample-efficiency. In summary, we see that CWM (without slots) performs well on the new compositional generalization experiments when given sufficient data.
>
>
> We then assessed the downstream task performance of our pretrained models. Here we constructed two new tasks - one control task and one prediction task. In the control task, we train a Soft Actor Critic (SAC) agent to manipulate a randomly sampled sprite in the Multi-dSprite environment to go to a particular location on the grid. The SAC agent receives observations that are the encodings of the scene produced by one of the pretrained models. We evaluate the agent using the embeddings of CSWM, CWM and the auto-encoder, and see that the agent trained to perform control using CSWM representations performs the best, with the CWM-based agent trailing closely behind (see Figure 9, page 15). This suggests that slotted representation could offer advantages in control tasks like this.
>
> In the second downstream task we used the representations of the trained MOVi-A models to predict a novel quantity. We froze the encoders of CSWM and CWM and trained a linear classifier to predict the number of objects present in a scene. Here we see that both models can predict object cardinality better than chance with a simple linear classifier with only minor differences in prediction accuracy between them (see Figure 10, page 16). Moreover, prediction accuracy generally increases the more data the models were trained with in their original task.
>
> Lastly, we followed the reviewer’s suggestion to further validate our forward accuracy metrics by training stop-gradient decoders to reconstruct images from latent states. In the Cubes and Multi-dSprite environments we trained CNN decoders for 100 epochs to reconstruct images from the representations of CSWM and CWM, respectively. We then evaluated the reconstruction accuracy with the LPIPS metric as the dynamics models predicted future states in an open-loop fashion. Matching our other prediction accuracy metric, we see that CWM retains lower LPIPS for future state predictions than CSWM in both environments. See Figure 11, page 16, and Figure 12 page 17 for example reconstructions and LPIPS scores for both environments.
>
> Questions:
> > The overview of object-centric representations is focused entirely on slot-based models.
>
> We thank the reviewer for this suggestion and have added the following paragraph in the related works section:
> **"While we focus on slot-based models, there are different approaches to learning object-centric representations. Patch-based models such as SPACE (Lin et al., 2020) and MarioNette (Smirnov et al., 2021) decompose scenes into disentangled representations by reconstructing the input from patches. Keypoint-based models such as DLP (Daniel & Tamar, 2022) build on representations as sets of geometrical points as an alternative to single-vector representations."**
>
>
> > What does it mean for the prediction to be “closest to the last frame”, closest compared to what?
>
> We evaluate whether the predicted state is the closest to the last frame in the sequence out of all predicted states in the. This means that, in a test set of a 1000 videos, we compare the final encoded state with the predicted final states for all 1000 videos. The prediction is deemed correct if the final state is the closest (in terms of L2 distance) to the predicted final state in the corresponding video, and incorrect if it is closer to any of the other 999 predictions. We compute this for all videos in the test set and report the percentage of correct predictions. We added the following clarification on line 268:
>
> **"In other words, a prediction is deemed correct if the final encoded state is the closest in $L_2$ distance to the predicted final state in the corresponding video, and incorrect if it is closer to any of the other 999 predictions."**

---

> > ### Author Response · Authors · 2024-11-21
> > **Part 2**
> >
> > >how do you compute the distance between two latent representations that are not from the same trajectory
> >
> > We compute distance the same way as in the original CSWM paper, by concatenating the slots together to form one joint representation of the scene, and calculating the  L2 distance between the predicted and encoded representation. This is part of the loss function of CSWM, encouraging it to learn aligned slot representations.
> >
> > >The original CSWM paper seems to have results on the cubes dataset that contradict your results
> >
> > We were also surprised to see that a model without slots could attain such high prediction accuracy on these datasets. The only changes we made to CSWM is replacing the last CNN layer that maps to object slots with a generic CNN layer, and replacing their GNN based transition dynamics module with an MLP with two hidden layers. We also kept the training setup and hyperparameters for our experiments.
> >
> > >  Is it possible that this is due to the fact that the auto-encoder was trained on that task precisely while the sequential auto-encoder was trained on next-frame prediction?
> >
> > Thanks for raising this point. This is indeed a possible explanation for this and we have updated the text to reflect this (line 377).
> >
> > **"Lastly, the auto-encoder performs better than the sequential auto-encoder on the test set, which might be explained by it having an extra objective in the loss function."**
> >
> > > For better qualitative assessment I would suggest adding additional metrics such as pixelwise MSE, SSIM, FID.
> >
> > Thanks for the suggestion. We have included a figure assessing reconstruction using both MSE and SSIM (Figure 17 and 18, page 19).
> >
> > >Is there a systematic separation between the train and test data in terms of compositions of objects and/or their properties such that performing well on the test data would necessarily require compositional generalization?
> >
> > There are no systematic differences in the train and test distributions of the datasets we trained the models on apart from the test set containing combinatorially novel data points (objects in novel combinations of positions, additionally with novel combinations of sizes/rotations in the dSprites environment, and novel colors and textures in MOVi-A). This is why we followed the reviewer’s suggestion to test CWM and CSWM in a setting where the train and test data differs in a more systematic way.
> >
> > Overall, we are very grateful for all the thoughtful suggestions and ideas for new experiments. We believe that conducting these has substantially improved the quality of our paper, and further strengthened the conclusions we make.

---

> > > ### Comment · Reviewer_xykj · 2024-11-23
> > >
> > > I would like to thank the authors for their effort in conducting additional experiments, answering questions and addressing the points that were raised by myself and other reviewers. I agree with the authors that these have improved the quality of the paper.
> > >
> > > While I genuinely find the subject of this study interesting and potentially useful, my main concerns remain mostly unchanged: (1) Some conclusions are not convincingly backed-up by experiments. (2) The implications of this study to downstream task performance are not thoroughly assessed or discussed.
> > >
> > > I now provide details explaining the reasons for my remaining concerns.
> > >
> > > **Learning Object Dynamics**
> > >
> > > The concerns I have here are regarding: (1) the breadth of the study with respect to types of models and (2) the metrics used to evaluate them.
> > >
> > > The first and major concern I have here, as I mentioned in my initial comment, is that the conclusions here are made based on a comparison between CWM and CSWM alone. In my opinion, this is not enough in order to make strong and convincing empirical conclusions. Experiments with at least the sequential auto-encoder compared to its slotted counterpart (e.g. Slotformer) are clearly missing, but the study could benefit from including more models from the contrastive and auto-encoding families.
> > >
> > > I thank the authors for clarifying the meaning of the *prediction accuracy* metric used in this section. I would define this metric a *self-consistency* measure rather than an *accuracy* measure. Self-consistency is required but not sufficient to conclude that the model indeed accurately predicts forward dynamics, nor that it generalizes compositionally. For this reason, I suggested training a stop-gradient decoder and using perceptual metrics on the actual images. While the authors have done this on some of the datasets, this assessment on the other (possibly more complex) datasets is still missing, as well as in the systematic compositional generalization experiments added by the authors during the rebuttal period (Appendix A.1). This study and visualization would also be more simple with the auto-encoding models.
> > >
> > > Further questions in this subject:
> > > - Can the authors please provide a possible explanation for the discrepancy with original C-SWM on the cubes dataset?
> > > - Which decoding mechanism did you use to produce the images from CSWM slots? Similar to the decoder in Slot Attention?
> > > - How many slots did you use for CSWM on each dataset?
> > > - The decoding results of CSWM on the Multi-dSprite look very bad, even for the single-step case. Could this be related to the previous two questions?
> > >
> > > **Benefits of Partially Entangled Representations**
> > >
> > > The concerns I have here are regarding: (1) the breadth of the study with respect to types of models and (2) evaluating the implications of this finding.
> > >
> > > Conclusions here are also made based on a comparison between CWM and CSWM alone. While the analysis is interesting in itself, its implications are not entirely clear. The authors claim that entangled representations “can also give rise to systematic representations of transformations that act on objects”, but e.g. for sequential decision-making which requires acting on objects, this does not prove to be beneficial based on your experimental results in Figure 9. This might suggest that this analysis is either not sufficient to make conclusions about systematic representations of transformations or maybe that cosine-similarity favors the single-vector representation for some reason and does not provide the full picture. In any case, additional experimental results including additional models and possible similarity measures are missing here.
> > >
> > > **Downstream Decision-Making**
> > >
> > > I thank the authors for making the effort and directly implementing my suggestions for downstream task performance. While these results shed light on some aspects of your study, they are barely discussed, nor is their connection to the various measures you propose such as linear separability and systematic action representation. In addition, they are not referenced at all in the main text.
> > >
> > > Further questions in this subject:
> > > - Do the different accuracy plots in Figure 10 refer to different datasets with a different variation in number of objects? This is not clear and some explanation should be added in either the related text or the figure caption.
> > > - What is the policy architecture you used for SAC? What architectures do you use for the classifiers? Makes sense to use a Transformer or a GNN for the CSWM policy and classifier as was used for the dynamics prediction model. If this was not the case, I would expect this change to further increase the gap in performance in favor of the slotted CSWM.

---

> > > > ### Comment · Reviewer_xykj · 2024-11-23
> > > >
> > > > **Summary**
> > > >
> > > > Trying to pinpoint what I believe is missing throughout your study, I would focus on these main points:
> > > > - Breadth of experimental evaluation with respect to model classes (contrastive, auto-encoding, etc.) and types (single-vector vs. slot representations).
> > > > - Grounding of proposed measures (linear separability, systematic action representation) to (downstream) performance. Without this correspondence convincingly demonstrated, the usefulness of these measures for making general insights, as intuitive and reasonable as they may seem, is questionable.
> > > >
> > > > *A final note regarding additional experiments*: while the effort in producing these is greatly appreciated, simply adding them in the Appendix without context is somewhat lacking. I would expect these results to be discussed and incorporated in the relevant sections of the main text or at least referenced in the main text.

---

> > > > > ### Author Response · Authors · 2024-11-25
> > > > >
> > > > > Thank you for the detailed response to our rebuttal!
> > > > >
> > > > > We believe the main contribution of our work is to show that even simple latent dynamics models (contrastive and reconstruction-based) can learn separable object representations and generalize about object dynamics in novel scenarios (even when there are systematic train-test differences as suggested by the reviewer). While we agree that further comparisons can be made with respect to downstream tasks, additional baselines and methods, we also believe that the experiments we have done, many of which were suggested by the reviewer, do a good job of add establishing what we want to show.
> > > > >
> > > > > We have added discussions of the new results to the experiments the reviewer suggested to the paper (see section 4). In particular, we added a discussion on the downstream task results, clarifying that slotted representations can be beneficial for control because they facilitate object-centric learning. We also address the reviewer's questions in the paper text:
> > > > >
> > > > > >Do the different accuracy plots in Figure 10 refer to different datasets with a different variation in number of objects? This is not clear and some explanation should be added in either the related text or the figure caption.
> > > > >
> > > > > We have added the following clarification in the main text:
> > > > > **"We constructed three datasets, where the number of possible objects present in a scene increased from two to four. In the first dataset there were therefore two possible labels (does the scene contain one or two objects?), and in the last dataset four labels (does the scene contain one, two, three or four objects).
> > > > >
> > > > > > What is the policy architecture you used for SAC? What architectures do you use for the classifiers? Makes sense to use a Transformer or a GNN for the CSWM policy and classifier as was used for the dynamics prediction model. If this was not the case, I would expect this change to further increase the gap in performance in favor of the slotted CSWM.
> > > > >
> > > > > We use a standard MLP architecture to implement the SAC policy, and concatenate object slot representations since CSWM learns aligned object slots. We added the following clarification in the Appendix (line 891).
> > > > >
> > > > > **"To implement the policy we use a standard MLP mapping latent representations to actions. For CSWM we concatenate object slots before passing it to the policy network since it learns aligned, temporally consistent object slots. One could replace the MLP with a Transformer or a GNN to potentially attain higher performance"**
> > > > >
> > > > > Thanks again for the suggestions.

---

> > > > > > ### Comment · Reviewer_xykj · 2024-11-25
> > > > > >
> > > > > > I thank the authors for their response. The authors have responded to some of my questions/points and disregarded others for reasons I do not understand. In any case, I feel they have made a genuine effort in addressing some of my concerns although they have mostly not been resolved.
> > > > > >
> > > > > > To clarify, I do not agree that the experiments you have conducted establish what you want to show (see details in my previous responses).
> > > > > >
> > > > > > I find the subject of this study, the proposed measures and the conducted experiments interesting, but not sufficiently grounded. This is an empirical study and as such, in my opinion, would require stronger empirical evidence convincing readers of the general relevance of the proposed metrics as well as of the conclusions suggested in this paper. The summary I provided in my previous comment distills what I find is still missing.
> > > > > >
> > > > > > Therefore, I cannot recommend acceptance of this paper in its current state, and leave my score unchanged.

---

### Official Review · Reviewer_BkHi · 2024-11-04

**Soundness:** 3
**Presentation:** 3
**Contribution:** 3
**Rating:** 6
**Confidence:** 2

**Summary:**

The paper studies object disentanglement in the representation of visual scenes. Specifically, object-centric representations enforce an inductive bias for disentangling representation of different objects.  The claims that (1) distributed representations (not object disentangled) may outperform object-centric  ones in down-stream tasks; (2) Limited object disentanglement (linear separability)  may arise even without forcing it through the architecture.

The paper compares the representations learning in structured and non-structured representations and analyzes their linear separability.

**Strengths:**

-- Studying inductive biases for neural architecture is interesting.
Its great to see a paper that is about more than just improving a few points on a known task.

-- Interesting experiments.

**Weaknesses:**

W1. There is a lot of work on compositional representations in machine vision.
From early papers focused on compositionality of attributes
 [Yuval Atzmon et al 2016, Justin Jhonson 2017].

W2. The main higlighted claim is that, ith sufficient data, models can learn representation that disentangle objects.
I am missing an argument that explains why that is surprising or unsexepected. Does theory sugests otherwise?
In attribute compositionality, it was argued that discriminative models entangle peoperties that are correlated intheir trainin data,
and cannot generalize to new combinations. But here, objects are not entangked dugin training. So why would disentaglement be a problem?

W3. The paper measures linear separability. This is a different concept that compositionality, and the paper should make the distinction super clear and explicit, already in the title.  In the ML literature, compositionality usually means that one can represent new things using combination of concepts.

W4. In high enough dimensions, it is easy to get linear separability, depending on the number of classes.

**Questions:**

Q1. What specific properties of the architecture or training procedure lead to linear separability effect?

Q2: How general are the results in the paper? Would they generalize to other object-centric approaches beside CWM / CSWM?

---

> ### Author Response · Authors · 2024-11-21
>
> We thank the reviewer for their review and engaging with our work. We are happy that the reviewer found our paradigm interesting. The reviewer also highlighted some important points, especially regarding the definition of compositionality and whether our metric is trivially maximized by models with high-dimensional latent spaces. We sought to address these concerns by showing that CWM can generalize compositionally about i) scenes with more objects than it was originally trained on, and ii) about objects with different combinations of attributes than trained on (such as objects with swapped colors). We further show that our object-separability metric cannot be maximized simply by having large feature spaces. A randomly initialized network with 2000 latent dimensions shows only slightly better than random object separability, whereas a trained model with 50 latent dimensions is at ceiling. We believe that addressing these points has greatly improved our manuscript. We address each point in more detail below.
>
> >There is a lot of work on compositional representations in machine vision. From early papers focused on compositionality of attributes
> We thank the reviewer for highlighting this.
>
> We have expanded the related work section with the following paragraph to include earlier papers on compositionality of attributes, as well as additional model types suggested by Reviewer xykj:
>
> **"Atzmon et al. (2016) and Johnson et al. (2017) introduced datasets to test compositional generalization in machine learning models. While we focus on slot-based models, there are different approaches to learning object-centric representations. Patch-based models such as SPACE (Lin et al., 2020) and MarioNette (Smirnov et al., 2021) decompose scenes into disentangled representations by reconstructing the input from patches. Keypoint-based models such as DLP (Daniel & Tamar, 2022) build on representations as sets of geometrical points as an alternative to single-vector representations."**
>
> > In attribute compositionality, it was argued that discriminative models entangle peoperties that are correlated intheir trainin data, and cannot generalize to new combinations. But here, objects are not entangked dugin training
>
> We thank the reviewer for raising this point. We conducted a new experiment in which CSWM and CWM were trained on object dynamics where objects were entangled with their attributes (cubes were always red and spheres always green), and tested on videos of objects where these attributes were flipped. Again we see that the models can generalize about the novel test combination of objects and attributes when given enough data (see Figure 8, page 15).
>
> > In high enough dimensions, it is easy to get linear separability, depending on the number of classes.
>
> This is a good point. In high enough dimensions linear separability could be trivial. This is why we install an L1 norm penalty on the linear classifier weights when we perform the linear separability analysis. We further tested whether models with high-dimensional latent spaces trivially attained high separability scores. We randomly initialized image encoders with 50, 100, 500, 1000, and 2000 latent dimensions, respectively. Without training, these models perform only slightly better than chance levels, whereas a trained CWM model with 50 latent dimensions is close to ceiling performance on our separability metric (see Figure 13, page 17).
>
> >What specific properties of the architecture or training procedure lead to linear separability effect?
>
> We find two factors important for separability: Training set size, and next-state prediction. We further see that the dynamic contrastive models (CWM) generally attain higher object decodability scores than auto-encoding models. We show the combined scores for all models in Figure 15, page 18.
>
> > How general are the results in the paper? Would they generalize to other object-centric approaches beside CWM / CSWM?
>
> We additionally show in the paper that models trained with auto-encoding objectives develop separable representations of objects when trained to predict dynamics. This suggests that this phenomenon might also generalize to other architectures and training setups.

---

> > ### Comment · Reviewer_BkHi · 2024-11-25
> > **Further discussion**
> >
> > Thank you for your response.
> >
> > (1) "Experiment with objects entangled with their attributes". I was actually aiming at something more basic.
> > My point was that in attribute entanglement, it is hard for models to disentangle two things that always appear together.
> > The parallel phenomenon for object disentanglement would be two object that always appear and move together.
> > For attributes, if attributes in the data appear in a disentangled way (e.g., all shapes come with all colors), models shouldn't find it very hard to learn a disentangled representation.
> >
> > The parallel here is that if objects appear and move in an uncorrelated way, I am not surprised that the representation they learned is disentangled. Why is it impressive or important that the model learned a representation that is disentangled?  This appears to be a property of the synthetic data.
> >
> > (2) I don't see a response to my W3 comment. (The paper measures linear separability. This is a different concept that compositionality, and the paper should make the distinction super clear and explicit, already in the title)

---

> > > ### Author Response · Authors · 2024-12-01
> > >
> > > Thanks for the response and clarifications!
> > >
> > > > Why is it impressive or important that the model learned a representation that is disentangled? This appears to be a property of the synthetic data.
> > >
> > > It is indeed true that the object trajectories are uncorrelated, potentially facilitating object-centric representation learning for the non-slotted models. We believe it's still surprising that unregularized latent dynamics models learn disentangled representations of objects: In [1], where a $\beta$-VAE is compared against unregularized alternatives, the authors show significantly better disentanglement with the same separability metric when training on a synthetic dataset with uncorrelated factors. Our results, on the other hand, show that training dynamics models on object trajectories is indeed sufficient to attain close to perfect object-disentanglement (again, using their metric), without regularization.
> > >
> > > > This is a different concept that compositionality, and the paper should make the distinction super clear and explicit, already in the title)
> > >
> > > Thanks for the suggestion. Disentanglement is closely related to compositionality, as we show that the latent dynamics models learn to decompose the scenes into separable representations that can be combined in a factorial manner. However, we take the reviewer's comment to heart and propose to change to the following alternative title "Next state prediction gives rise to entangled, yet separable representations of objects".
> > >
> > > We thank the reviewer for the fruitful discussion.
> > >
> > > [1] Higgins, Irina, et al. "beta-vae: Learning basic visual concepts with a constrained variational framework." ICLR (Poster) 3 (2017).

---

### Author Response · Authors · 2024-11-21
**General response**

We would like to thank all reviewers for their constructive interactions. The reviewers generally found our work novel and well written:
* Reviewer BkHi wrote “Its great to see a paper that is about more than just improving a few points on a known task.”
* Reviewer xykj said that our paper was “well written” and “novel”
* Reviewer ZxZn stated that “The question tackled by the authors is very interesting and has not been extensively investigated in the past.”
* Reviewer v3zg wrote that our paper “provides robust evidence across diverse datasets”


At the same time reviewers raised some important points. In general, reviewers suggested several experiments and analyses that could be conducted to further strengthen the claims made in the paper. To address these points, we conducted extensive additional experiments showing

* How non-slotted models can perform compositional generalization when there are systematic differences in the train-test splits, or when object attributes are entangled (Figure 8, page 15).

* How well the models perform in two new downstream tasks (Figure 9 and 10, page 15-16)

* That image reconstruction can be done from the converged contrastive models  (Figure 11 and 12, page 16-17)

* That having a high-dimensional representational space alone is not sufficient to attain a high object separability score (Figure 13, page 17).

The results for all of these experiments can be viewed in the Appendix of the updated paper, starting from page 15. Changes are marked in red. We describe these and other smaller changes in detail in our responses to the individual reviews below. We would like to thank the reviewers again for their valuable input, which has been indispensable for improving our paper.

---

### Meta-Review · Area_Chair_yKvm · 2024-12-19

**Metareview:**

This paper studies structured (object-centric) vs. unstructured visual representations in the context of next-state prediction. While the paper is well written and tackles a very interesting problem, the reviewers voiced mixed opinions about the strength of the experimental evaluation of the claims made in the paper. The AC agrees with these concerns and encourages the authors to resubmit an improved version of the paper to a later venue.

**Additional Comments On Reviewer Discussion:**

The discussion period did not change the overall view of the paper and no reviewer was willing to champion acceptance.

---

### Decision · Program_Chairs · 2025-01-22

Reject